# Identification of CircRNA signature associated with tumor immune infiltration to predict therapeutic efficacy of immunotherapy

Yu Dong[1,2,3,4,14], Qian Gao[1,5,6,14], Yong Chen [7,8,14], Zhao Zhang[9,10,14], Yanhua Du[2,3,14], Yuan Liu [9,11,12], Guangxiong Zhang[4,5], Shengli Li [13], Gaoyang Wang [2,3], Xiang Chen [1,5] ✉, Hong Liu [1,5] ✉, Leng Han [9,11,12] ✉ & Youqiong Ye [2,3] ✉

Circular RNAs (circRNAs) play important roles in the regulation of cancer. However, the clinical implications and regulatory networks of circRNAs in cancer patients receiving immune checkpoint blockades (ICB) have not been fully elucidated. Here, we characterize circRNA expression profiles in two independent cohorts of 157 ICB-treated advanced melanoma patients and reveal overall overexpression of circRNAs in ICB non-responders in both pre-treatment and early during therapy. Then, we construct circRNA-miRNA-mRNA regulatory networks to reveal circRNA-related signaling pathways in the context of ICB treatment. Further, we construct an ICB-related circRNA signature (ICBcircSig) score model based on progression-free survival-related circRNAs to predict immunotherapy efficacy. Mechanistically, the overexpression of ICBcircSig circTMTC3 and circFAM117B could increase PD-L1 expression via the miR-142-5p/PD-L1 axis, thus reducing T cell activity and leading to immune escape. Overall, our study characterizes circRNA profiles and regulatory networks in ICB-treated patients, and highlights the clinical utility of circRNAs as predictive biomarkers of immunotherapy.

[1]Department of Dermatology, Hunan Key Laboratory of Skin Cancer and Psoriasis, Hunan Engineering Research Center of Skin Health and Disease, Xiangya Clinical Research Center for Cancer Immunotherapy, Furong Laboratory, Changsha, Hunan 410008, P. R. China. [2]Center for Immune-Related Diseases at Shanghai Institute of Immunology, Ruijin Hospital, Shanghai Jiao Tong University School of Medicine, Shanghai 200025, P. R. China. [3]Shanghai Institute of Immunology, State Key Laboratory of Oncogenes and Related Genes, Department of Immunology and Microbiology, Shanghai Jiao Tong University School of Medicine, Shanghai 200025, China. [4]Lin Gang Laboratory, Shanghai 200025, China. [5]National Clinical Research Center for Geriatric Disorders, Xiangya Hospital, Changsha, Hunan 410008, P. R. China. [6]Department of Clinical Laboratory, Xiangya Hospital, Central South University, Changsha, Hunan, China. [7]Department of Musculoskeletal Surgery, Fudan University Shanghai Cancer Center, Shanghai 200032, P. R. China. [8]Department of Oncology, Shanghai Medical College, Fudan University, Shanghai 200032, P. R. China. [9]Department of Biochemistry and Molecular Biology, McGovern Medical School at The University of Texas Health Science Center at Houston, Houston, TX 77030, USA. [10]MOE Key Laboratory of Metabolism and Molecular Medicine, School of Basic Medical Sciences, Fudan University, Shanghai 200433, P. R. China. [11]Center for Epigenetics and Disease Prevention, Institute of Biosciences and Technology, Texas A&M University, Houston, TX 77030, USA. [12]Department of Translational Medical Sciences, College of Medicine, Texas A&M University, Houston, TX 77030, USA. [13]Precision Research Center for Refractory Diseases, Institute for Clinical Research, Shanghai General Hospital, Shanghai Jiao Tong University School of Medicine (SJTU-SM), Shanghai 201620, China. [14]These authors contributed equally: Yu Dong, Qian Gao, Yong Chen, Zhao Zhang, Yanhua Du. ✉e-mail: chenxiangck@126.com; hongliu1014@csu.edu.cn; leng.han@tamu.edu; youqiong.ye@shsmu.edu.cn

Melanoma is the most common histological subtype of skin cancer and causes approximately 75% of deaths related to skin cancer with, an incidence of 15–25 per 100,000 individuals at the global level[1]. The median survival time of metastatic melanoma patients is just 6–12 months[2]. Immune checkpoint blockade (ICB) therapies targeting programmed cell death receptor 1(PD-1) and cytotoxic T lymphocyte antigen 4 (CTLA-4) have been a revolutionary breakthrough in oncology, especially for the treatment of metastatic melanoma[3–6]. Unfortunately, only a small proportion of patients achieve durable clinical benefits from ICB immunotherapy[7,8]. It is thus urgent that additional predictive biomarkers be identified to guide immunotherapy utilization for precision oncology.

Circular RNAs (circRNAs) are a class of single-stranded noncoding RNA characterized by a covalently closed circular structure[9], generated from pre-mRNAs through the back-splicing of a downstream 5′ splice site to an upstream 3′ splice site[10]. CircRNAs are involved in various biological and cellular functions, such as tumorigenesis[11] and the epithelial-mesenchymal transition (EMT)[12], though their abilities to bind specific miRNAs[13] and/or proteins[14]. Notably, recent studies have demonstrated that circRNAs are participants in the regulation of various anti-tumor immune responses and immune cells[15]. For example, the circRNA *hsa_circ_0020397* can bind and inhibit the expression of miR-138, which targets PD-L1 to inhibit its expression. Therefore, the overexpression of *hsa_circ_0020397* promotes the upregulation of PD-L1, leading to immune escape in colorectal cancer[16]. CircRNAs may also interact with proteins. For example, *circFoxo3* can regulate p53-influenced immune responses by inducing the ubiquitination-dependent degradation of p53 through binding to MDM2[17,18]. Tumor cells may produce abnormal circRNAs caused by genetic mutations[19], chromosomal translocation[20], TGF-β signaling regulation[21], and other aberrant events[22]. For example, PML/RARα chromosomal translocations lead to the generation of fusion circRNAs (F-circRNAs) in acute promyelocytic leukemia, which promote cellular transformation, cell viability, and resistance to therapy[23]. Furthermore, there is evidence of a correlation between circRNAs and the infiltration of immune cells in several cancers[24,25]. These studies suggest that circRNAs play important roles in the tumor microenvironment (TME), which may further enable them to predict patient responses to immunotherapy[26]. Despite the emerging roles of circRNAs in the immune system, no studies have systematically profiled the circRNA expression landscape involved in cancer immunotherapy.

To date, several biomarkers associated with response to ICB treatment have been identified. Tumor mutational burden (TMB) (≥10 mutations/megabase) has been approved as an ICB therapeutic biomarker for the treatment of unresectable or metastatic solid tumors with pembrolizumab[27,28]. PD-L1 expression, as determined by immunohistochemistry (IHC), is also currently used clinically as a companion diagnostic biomarker, although such staining has been found to be an inadequate determinant of treatment benefit in multiple clinical trials[29–32]. CD8+ T cells, which are critical for tumor cell recognition and killing[33], have been identified as a positive biomarker to predict ICB responses in multiple cancer types[34]. Transcriptome-based signatures have also been proposed as candidate biomarkers of ICB responses[35–51], such as IMmuno-PREdictive Score (IMPRES)[47], interferon (IFN)-γ signaling pathways[52], and Innate anti-PD-1 Resistance (IPRes) Signatures[27]. However, most of these transcriptomic signatures are derived from a single tumor type and a limited number of patients, constraining their utility. Recently, circRNAs, a class of noncoding RNAs, have been identified as potential biomarkers associated with disease diagnosis and treatment[53]. However, there are no circRNA-based biomarkers for the therapeutic efficacy of ICB.

In this study, we characterized the expression landscape of circRNAs using total RNAseq data from two melanoma patient cohorts, including 88 melanoma patients treated with single-agent anti-PD-1 or combined anti-CTLA-4 and anti-PD-1 immunotherapy[54], and 69 melanoma patients treated with anti-PD-1 therapy[55]. Through this approach, we identified several differentially expressed circRNAs and demonstrated their associations with patient survival. Furthermore, we constructed a circRNA signature (ICBcircSig) score for predicting the efficacy of ICB through the use of a machine learning technique, and further validated this signature in an independent cohort. Our findings unveil the significance of aberrant circRNA expression in patients with ICB treatment and provide insight into the potential applications of circRNA signatures in ICB therapy.

## Results
### CircRNA profiling in two independent melanoma patient cohorts undergoing ICB treatment

To characterize the significant roles that circRNAs may play in differentiating between response and non-response in patients undergoing ICB treatment, we retrieved two independent total RNA sequencing datasets for individuals treated with single-agent anti-PD-1 and/or combined anti-CTLA-4 and anti-PD-1. Cohort 1 consisted of 88 patients treated with anti-PD-1 monotherapy (n = 47, including pre-treatment [PRE] samples, n = 38, and early during therapy [EDT] samples, n = 9) or combined anti-CTLA-4 and anti-PD-1 therapy (patients, n = 41, including PRE, n = 32 and EDT, n = 9; Supplementary Data 1), while Cohort 2 consisted of 69 melanoma patients undergoing anti-PD-1 therapy (nivolumab or pembrolizumab) (Supplementary Data 2). The unmapped reads per patient ranged from 1,646,487 to 28,592,124 reads in Cohort 1 and from 1,044,568 to 7,607,887 reads in Cohort 2 (Supplementary Fig. 1a, b), and these data were used to quantify circRNA expression. To reliably identify circRNAs, we combined four well-established circRNA-detection tools with user-friendly computational algorithms[21], including CIRI2[56], find_circ[57], CircExplorer2[58], and CircRNA_finder[59], to quantify back splice-spanning reads (Fig. 1a, b; See Methods). There was no significant correlation between the number of unmapped reads and the number of circRNAs detected by any of these circRNA-detection tools (Supplementary Fig. 1c, d), suggesting that the detectable number of circRNAs is not dependent on the number of unmapped reads. Different circRNA-detection tools identified varying numbers of circRNAs[60] (Fig. 1a, b), so we kept those circRNAs identified by at least two tools with ≥2 back-splicing reads. We ultimately identified 89,204 total circRNAs from the 88 samples in Cohort 1 (Fig. 1c, Supplementary Data 3), and 43,911 circRNAs from the 69 samples in Cohort 2 (Fig. 1d, Supplementary Data 4). The detectable number of circRNAs ranged from 1440 to 16,653 in each patient from Cohort 1 (Supplementary Fig. 1e), and from 39 to 11,652 in each patient from Cohort 2 (Supplementary Fig. 1f). To minimize potentially spurious events, we only considered circRNAs identified in more than 20% of the total samples in each cohort. A total of 5350 circRNAs were retained in Cohort 1, ranging from 774 to 4525 in each patient (Fig. 1e), while a total of 3654 circRNAs were retained in cohort 2, ranging from 20 to 3013 in each patient (Fig. 1f).

In patient Cohort 1, the 5350 identified circRNAs originated from 2678 host genes, with most of these host genes (1515/2678 = 56.6%) having at least one circRNA, while 16 host genes were associated with more than 10 circRNAs (Fig. 1g). In patient Cohort 2, the 3654 identified circRNAs originated from 2110 host genes, and 1333 host genes were associated with one circRNA, while 5 host genes were associated with more than 10 circRNAs (Fig. 1h). We found that circRNAs detected in this study were ubiquitously located across whole genomic regions (Supplementary Fig. 1g, h), and most circRNAs were back-spliced from exonic regions (Supplementary Fig. 1i, j). We performed overlap analyses among the two cohorts in this study, circAtlas[61], and the MiOncoCirc[62,63] database. A significant overlap of detectable circRNAs was observed between Cohort 1 and Cohort 2, with 90.4% (3293/3644) of the circRNAs in Cohort 2 having been detected in Cohort 1 (Fisher test, p < 2.2e-16; Fig. 1i). We further found that

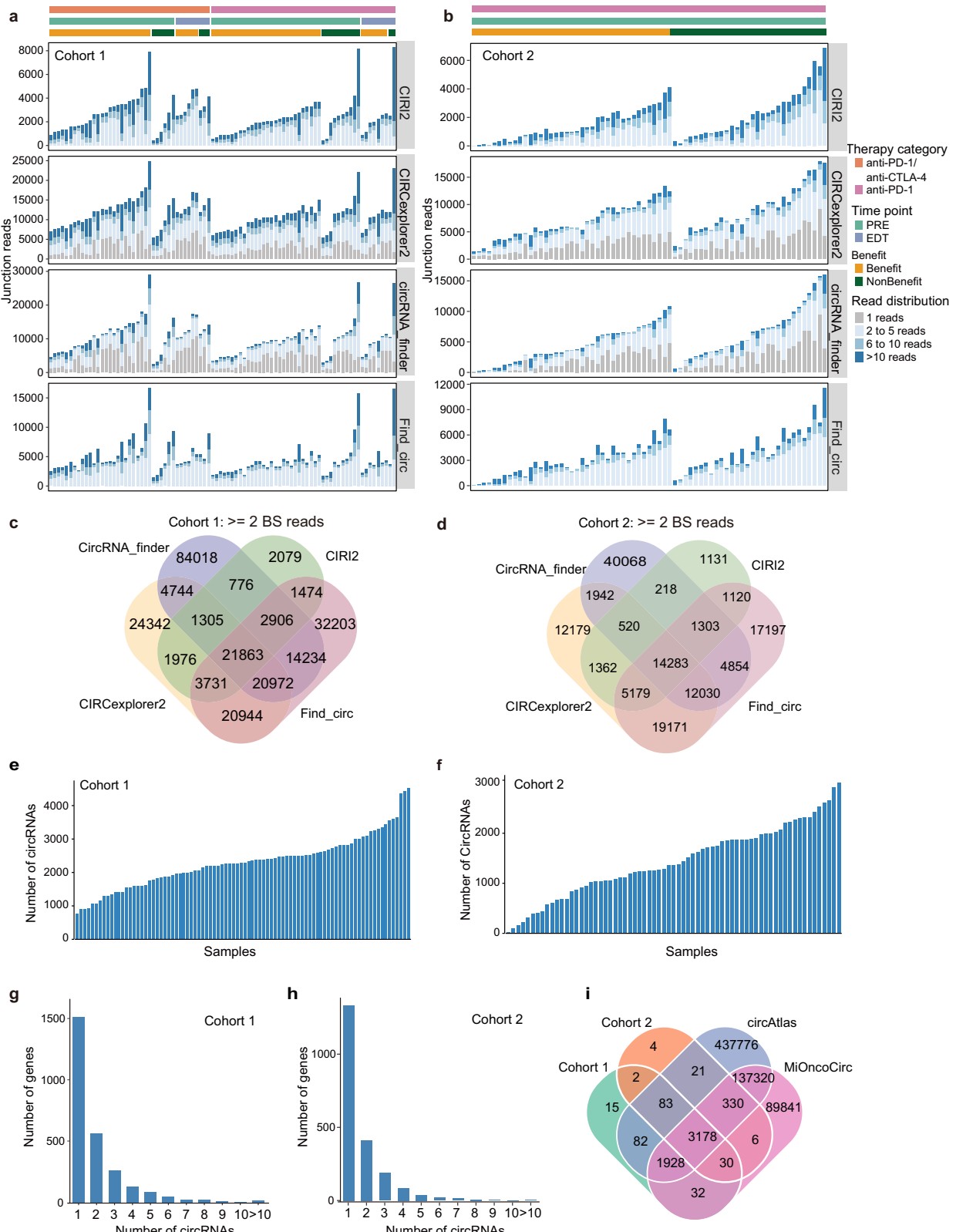

**Fig. 1 | Characterization of circRNAs in non-response and response group to ICB treatment in two cohort.** Combination of four computational tools to identify back-splice junction reads from total RNA sequencing data in melanoma patients with ICB treatment in cohort 1 (**a** *n* = 70) and cohort 2 (**b** *n* = 69). The x-axis indicates the tumor samples, sorted by treatment category, time point, treatment efficacy, and junction reads calculated by the four tools. Number of identified circRNAs for each tool in cohort 1 (**c**) and cohort 2 (**d**). Number of identified circRNAs for each sample and generated from host gene in cohort 1 (**e**, **g**) and cohort 2 (**f**, **h**). **i** The overlap of identified circRNAs between cohort 1, cohort 2 and public circRNA database cicrAtlas and MiOncoCirc. BS, back splice. Source data are provided as a Source data Fig. 1a, b, e–h.

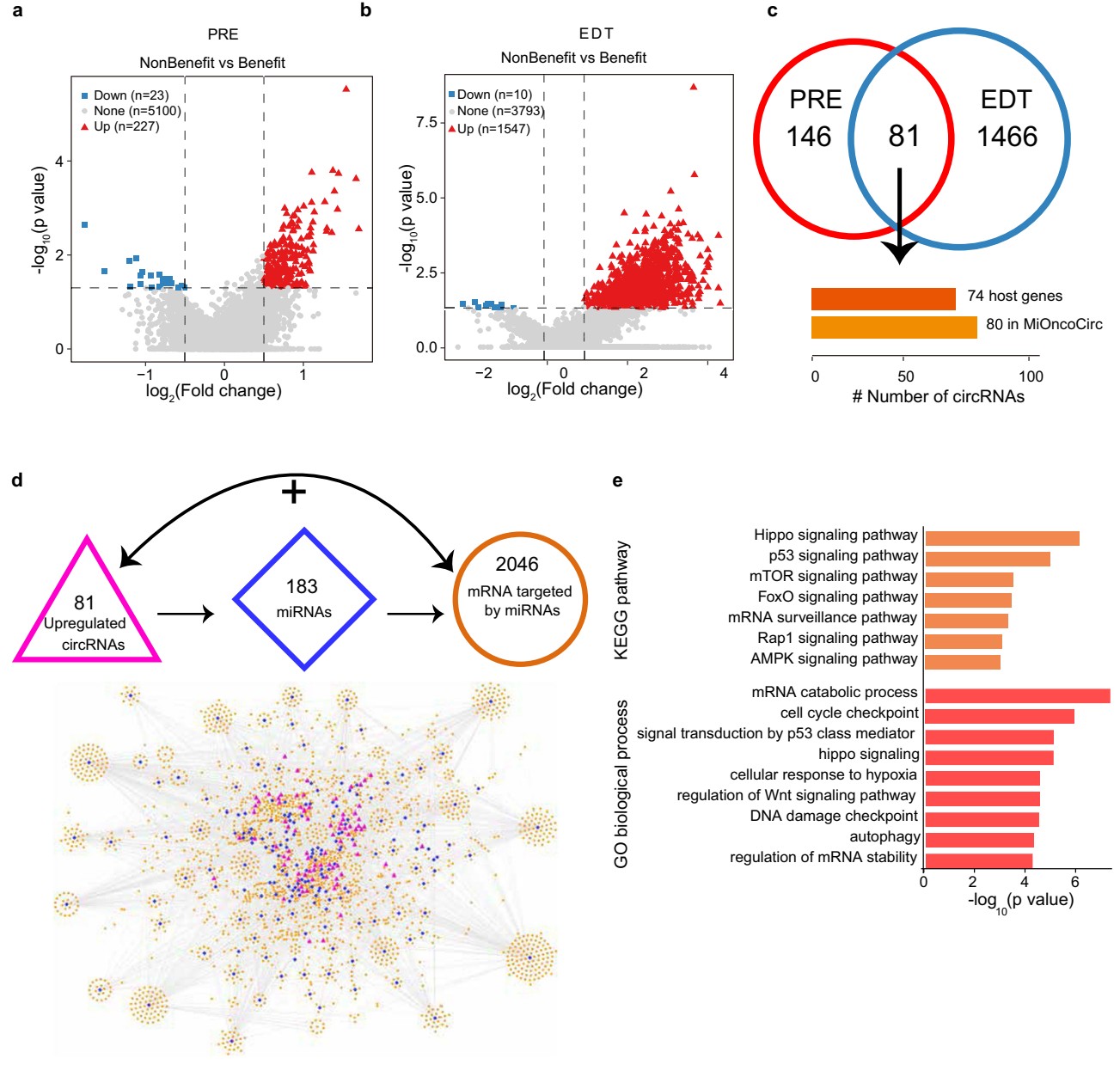

**Fig. 2 | Differentially expressed circRNAs between non-response and response patients and circRNA-miRNA-mRNA axes.** Volcano plot of upregulated (red) and downregulated (blue) circRNAs between non-response and response melanoma samples in pre-treatment (PRE; **a**) and early during therapy (EDT; **b**) samples, respectively ($P < 0.05$, |log2 (fold change)| ≥0.5; See Methods). **c** The overlap of upregulated circRNAs identified in PRE (**a**) and EDT (**b**) samples (81 shared circRNAs with 80 in MiOncoCirc and 74 host genes). **d** Strategy of ICB associated circRNA-miRNA-mRNA axes (upper panel, see method), and the network of all circRNA-

miRNA-mRNA interaction pairs (bottom panel). "+" represent the positive correlation between the expression of circRNAs and mRNAs. **e** Kyoto Encyclopedia of Genes and Genomes (KEGG) enrichment and GO biological process analysis among the mRNAs identified in ICB-associated circRNA-miRNA-mRNA axes ($P < 0.05$). The statistical analysis was performed by a two-sided linear mixed-effects model (LME) in **a**, **b** and two-sided Fisher's test in **e**. PRE pre-treatment, EDT early during therapy. Source data are provided as a Source data Fig. 2.

96.6% of the circRNAs (5168/5350) in Cohort 1 and 97.3% (3544/3654) in Cohort 2 were identified in MiOncoCirc database, suggesting circRNAs in these two cohorts have also been identified in various tumor tissues. 98.5% of the circRNAs (5271/5350) in Cohort 1 and 98.8% (3612/3654) in Cohort 2 were identified in the circAtlas database[61]. Furthermore, 95.4% of circRNAs (5106/5350) in Cohort 1 and 96% (3508/3654) in Cohort 2 were identified in both the circAtlas and MiOncoCirc databases, suggesting the conserved expression of these circRNAs in both normal and tumor samples.

## Identification of circRNAs associated with immune responses through circRNA-miRNA-mRNA regulatory axes

To better understand the molecular regulation of tumor responses to immunotherapy, we used a linear mixed effects (LME) model to identify circRNAs that were differentially expressed between the response ($n = 54$) group, defined as partial/complete response (PR/CR) or stable disease (SD) with progression-free survival (PFS) >3 months and the non-response ($n = 16$) group, defined as progressive disease (PD) or SD with PFS <= 3 months before treatment (PRE), with the expression of these differentially expressed circRNAs then being further examining

early during therapy (EDT) (response, $n = 13$; non-response, $n = 5$). In the PRE biopsies, we found 227 upregulated and 23 downregulated circRNAs ($P < 0.05$ and $|\log_2$ (fold change) $| \geq 0.5$) in non-responder patients compared to responder patients. In EDT biopsies, we detected 1547 upregulated and 10 downregulated circRNAs after ICB treatment (Fig. 2a, b). More upregulated circRNAs and fewer downregulated circRNAs were found in the non-responder group at both the PRE and EDT time points, suggesting a potential association between the overexpression of circRNAs and resistance to immunotherapy. CircRNAs were relatively highly abundant in the non-responder group, whereas no significant differences in the expression of their host genes were observed (Supplementary Fig. 2), suggesting the independent roles of circRNAs in the response to immunotherapy.

To further explore the functional effects of circRNAs in immunotherapy, we have assumed that circRNA expression at PRE time point may be elevated in order to be elevated at EDT time point, and thus focused on 81 upregulated circRNAs shared between non-responder and responder patients at the PRE and EDT timepoints for subsequent analysis (Supplementary Table 3), which were derived from 74 host genes (Fig. 2c). We focused on the circRNA-miRNA-mRNA interactions in which circRNA expression was positively correlated with mRNA expression to identify potential functional roles for these upregulated circRNAs in immunotherapy responses. We constructed circRNA-miRNA-mRNA regulatory networks mediated by common miRNAs that bind to the upregulated circRNAs and mRNAs and are correlated with the expression of those circRNAs. We used the miRanda algorithm[64] to predict high-confidence binding sites, and selected the top 30 miRNAs for each of 81 upregulated circRNAs. A total of 773 miRNAs and 2429 interactions were detected for the circRNA-miRNA axis. Next, we identified the potential targets of miRNA-mRNA interactions using the Tarbase databases[65], which incorporates experimentally supported interactions, and the TargetScan database. Taken together, these analyses enabled us to identify 184,587 predicted circRNA-miRNA-mRNA interactions based on the miRNA target sites shared by circRNAs and mRNAs. We filtered low-confidence associations based on multiple criteria including expression correlations (see Methods), after which 8449 (81 circRNAs – 183 miRNAs – 2046 mRNAs) ICB response-associated interactions were retained for further study (Fig. 2d; Supplementary Data 6).

Pathway analyses based on these 2046 mRNAs suggested their significant enrichment in cancer signaling pathways, including the Hippo, p53, mTOR, and AMPK signaling pathways (Fig. 2e. These mRNA targets were also enriched in several biological processes, including cell cycle checkpoint, cellular response to hypoxia, autophagy, and regulation of Wnt signaling pathways. For example, targeting Wnt/β-Catenin signaling may reverse the resistance to immunotherapy by altering antigen presentation[66]. Our analysis revealed the unexpected upregulation of circRNAs in non-responder patients, suggesting that certain circRNAs may mediate tumor resistance to immunotherapy through the alteration of cancer signaling pathways.

### Identification of an ICBcircSig prognostic model to predict immunotherapeutic efficacy

CircRNAs may serve as potential biomarkers in cancer[67], but it is unclear whether they can serve as biomarkers capable of predicting patient responses to cancer immunotherapy. To identify prognosis-related cirRNAs, we performed univariate Cox regression analyses examining the relationship between patient PFS and the expression levels of 227 upregulated circRNAs in PRE biopsies from Cohort 1. We found that high levels of expression for 25 circRNAs were significantly associated with poorer PFS (log-rank test, FDR < 0.05, and Cox FDR < 0.05). To identify the optimal circRNAs for use as prognostic biomarkers, we employed a LASSO Cox regression model analysis[68] using the expression profiles for these 25 circRNAs and associated clinical

information, and ultimately selected four circRNAs with non-zero regression coefficients (Supplementary Fig. 3a, b). We considered these four circRNAs as variables in a subsequent multivariate Cox regression analysis and found both *circTMTC3* and *circFAM117B* to be significant predictors of patient survival (Supplementary Fig. 3c). Specifically, the expression of these two circRNAs was associated with worse PFS (Fig. 3a, b), and patients that exhibited CR/PR or SD following anti-PD-1 therapy exhibited significantly lower expression of *circTMTC3* and *circFAM117B* as compared to patients with PD (Fig. 3c, d). We further collected 11 and 4 pre-treatment tumor biopsies from patients who did or did not respond to anti-PD1 treatment (Supplementary Table 1). Expression levels of *circTMTC3* and *circFAM117B* were detected in these samples by qRT-PCR. We found that both circRNAs were significantly upregulated in the non-responder group relative to the responder group (Fig. 3e, g). Sanger sequencing further confirmed that these PCR products spanned the circular junction for both *circTMTC3* and *circFAM117B* (Fig. 3f, h).

We further constructed an ICB-related circRNA signature (ICB-circSig) score by weighting the expression values of circRNAs in ICB-circSig based on their established multivariate Cox regression coefficient values. When assessing the clinical relevance of these scores, we found that the ICBcircSig score was able to distinguish between responder and non-responder patients with an AUC of 0.8 (Fig. 3i). We then determined that patients with a high ICBcircSig score had worse PFS compared with those patients with a low ICBcircSig score (log-rank test, $p < 0.001$, Fig. 3j). The 12 and 24-month progression rates in the high ICBcircSig score group were 100% and 100%, respectively, which were significantly higher than the rates (27% and 30%, respectively) in the low ICBcircSig score group. The respective AUCs of the time-dependent ROC curves for this ICBcircSig score were 0.76 and 0.75 for 12- and 24-month PFS (Fig. 3k). The predictive ability of the ICBcircSig score was validated through the random sampling of 90% of these samples 10 times, yielding mean time-dependent ROC curve AUC values for the ICBcircSig score of 0.757 and 0.754 for 12- and 24-month PFS (Supplementary Fig. 3d, e), respectively.

To further examine whether the ICBcircSig scores could serve as an independent prognostic factor in melanoma patients, we performed multivariate Cox regression analyses to adjust for the potential confounding effects of other conventional clinical factors, including age, gender, ICB treatment, CD274 (PD-L1), and PDCD1. The ICBcircSig score (hazard ratio [HR] = 2.975, 95% confidence interval [CI] 1.723–5.138, $p < 0.001$) remained an independent prognostic risk factor for PFS even after adjusting for these confounding factors (Fig. 3l). We further investigated the association between the ICBcircSig score and patient response to ICB treatment and demonstrated that responders exhibited significantly lower ICBcircSig scores as compared to non-responders (R vs NR, $p = 1.8 \times 10^{-4}$; Fig. 3m). In terms of overall survival, a higher ICBcircSig score was also associated with a shorter survival duration (log-rank test, $p = 0.028$, Fig. 3n).

### Validation of the performance of ICBcircSig prognostic model in an independent patient cohort

We further assessed the performance of the ICBcircSig score in an independent patient cohort. *CircTMTC3* and *circFAM117B* were associated with worse PFS and tended to be enriched in the patients that exhibited a PD response to ICB treatment in Cohort 2 (Supplementary Fig. 4a–d). We further calculated the ICBcircSig scores for each sample, and found ICBcircSig scores to be higher in the non-responder group (Fig. 4a), consistent with our observation in Cohort 1. ICBcircSig scores were also able to distinguish between responders and non-responders (AUC = 0.66; Fig. 4b). Survival analyses revealed that patients with high ICBcircSig scores exhibited worse PFS as compared to patients with low ICBcircSig scores (Fig. 4c), and the respective AUCs of the time-dependent ROC curves for the ICBcircSig score were 0.69 and 0.65 for 12- and 24-month PFS (Fig. 4d). The predictive ability of the ICBcircSig

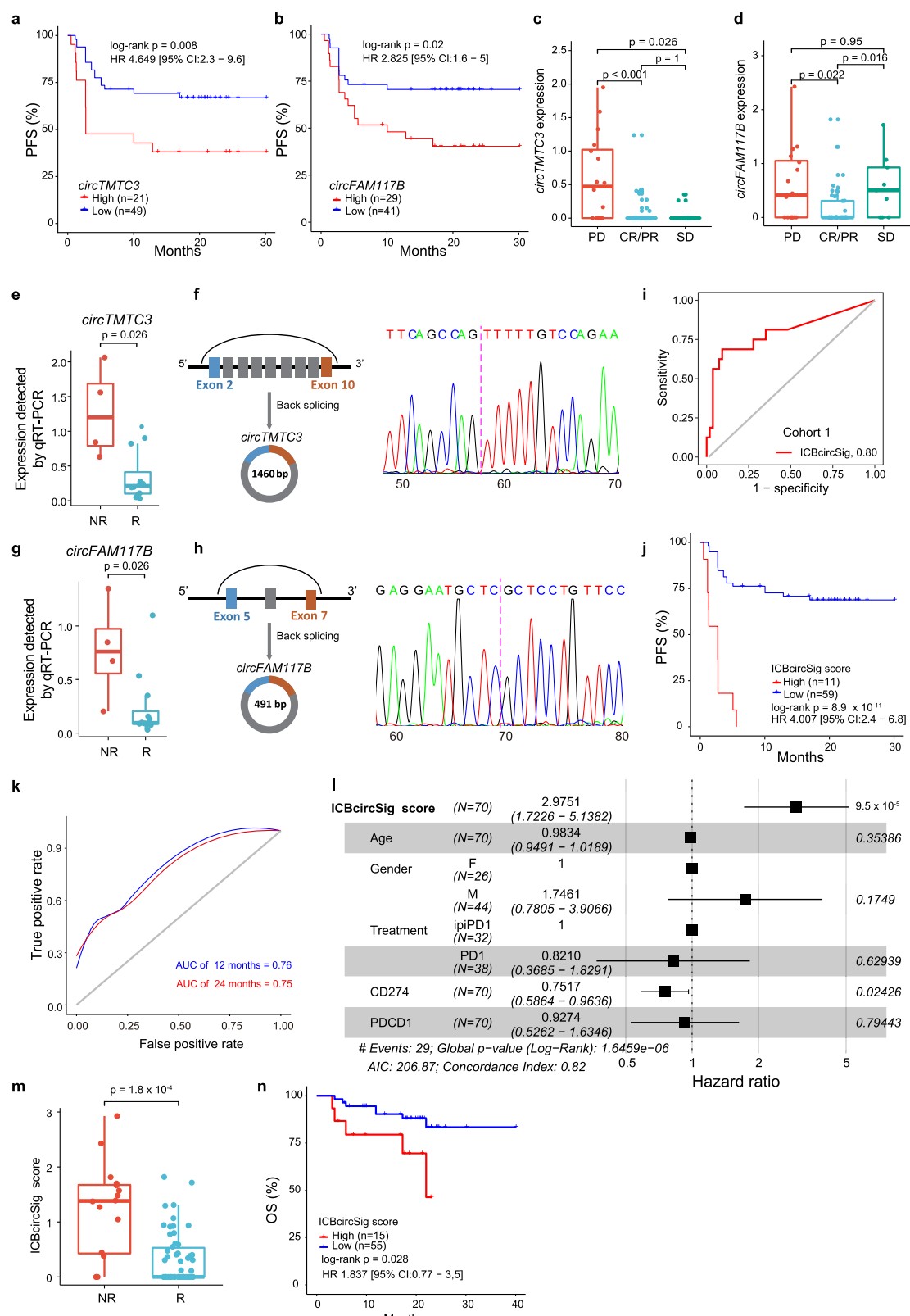

score was further validated via the random sampling of 90% of these samples 10 times, yielding respective mean AUC values for ICBcircSig scores of 0.678 and 0.645 for 12- and 24-month PFS (Supplementary Fig. 4e, f). To assess whether the ICBcircSig score is an independent predictor for PFS, we considered stage, gender, CD274 expression, PDCD1 expression, TMB, lactate dehydrogenase (LDH), and ICBcircSig scores as variables in a multivariate Cox analysis, and we ultimately

found that ICBcircSig score (HR = 1.32, 95% CI: 1.0721–1.630, $p$ = 0.009) was an independent predictor significantly associated with worse PFS (Fig. 4e). We further found that higher ICBcircSig scores were associated with worse overall survival (Fig. 4f), and these ICBcircSig score model exhibited strong prognostic performance, with respective AUC values for the ICBcircSig score of 0.71 and 0.67 for 12- and 24-month OS (Fig. 4g). ICBcircSig score remained an independent predictor of

**Fig. 3 | Construction and validation of ICBcircSig as the prognostic biomarker.** Kaplan–Meier survival curves show expression of *circTMTC3* (**a**) and *circFAM117B* (**b**) in the ICBcircSig associated with PFS. The expression of *circTMTC3* (**c**) and *circFAM117B* (**d**) expression were significantly higher in PD groups ($n = 16$) than CR/PR or SD groups ($n = 54$). **e–h** The qRT-PCR and Sanger sequencing validate expression and junction of circRNAs. The expression of *circTMTC3* (**e**) and *circFAM117B* (**g**) were detected by qRT-PCR in pre-treatment tumor biopsies from patients show response ($n = 11$) and non-response ($n = 4$) to anti-PD-1 therapy. The schematic diagram show generation of *circTMTC3* (**f**, left panel) and *circFAM117B* (**h**, left panel). Sanger sequencing confirmed that PCR products spanned the circular junction of predicted *circTMTC3* (**f**, right panel) and *circFAM117B* (**h**, right panel) in the patient with non-response to anti-PD-1 therapy. The magenta dash line indicates the circular junction site. **i** ROC curves quantifying ICBcircSig score response prediction AUC in cohort 1. **j** Kaplan–Meier survival curves of PFS between high- and low-risk patients stratified by ICBcircSig score using the optimal cutoff. **k** Time-dependent ROC curve at 12 and 24-months of PFS for the ICBcircSig score. **l** Forest plot for the HRs of multivariate Cox model of the ICBcircSig score and clinicopathological variables. Black vertical lines indicate the 95% confidence interval (CI). **m** Boxplot of ICBcircSig score among NR ($n = 16$) and R ($n = 54$) groups. **n** Kaplan–Meier survival curves of OS between high- and low-risk patients stratified by ICBcircSig score. A log-rank test was used in **a**, **b**, **j**, and **n**. A two-sided Wilcoxon rank-sum test was used in **c–e**, **g**, and **m**. The box in **c–e**, **g**, and **m** showed the median ± 1 quartile, with the whiskers extending from the hinge to the smallest or largest value within 1.5× IQR from the box boundaries. PFS progressive free survival, OS overall survival, CR/PR complete response/partial response, SD stable disease, PD progressive disease, ROC receiver operating characteristic curve, HR hazard ratio, R responder, NR non-responder. Source data are provided as a Source data Fig. 3a–e, g, i–k, m, n.

melanoma patient OS even after adjusting for confounding factors (Fig. 4h).

In addition, we validated the ICBcircSig score using our in-house cohort (Cohort 3) consisting of 24 patients with melanoma undergoing anti-PD-1 treatment (Supplementary Table 2, Supplementary Data 7). Consistently, *circTMTC3*, *circFAM117B*, and ICBcircSig score values were significantly enriched in non-responders (Fig. 4I, j). In this cohort, the ICBcircSig score exhibited an AUC of 0.85 when predicting patient responses to ICB treatment (Fig. 4k). A higher ICBcircSig score was associated with worse PFS (Fig. 4l), and the AUCs of time-dependent ROC curves for the ICBcircSig score at 12 and 18-months were 0.82 and 0.76, respectively (Fig. 4m). Taken together, our results provide proof-of-concept evidence that this ICBcircSig score may accurately predict patient responses to immunotherapy.

## The ICBcircSig score outperforms other transcriptome-based signatures

Previous studies have explored multiple transcriptome-based signatures to predict patient response to ICB treatment[35–51]. We next compared the performance of the ICBcircSig score with 20 previously reported signatures (Supplementary Table 3) to assess the predictive ability of these tools in our patient cohorts. The ROC classification curves for non-responders and responders based on the ICBcircSig score yielded AUC values of 0.80, 0.66, and 0.85 for Cohort 1, Cohort 2, and our in-house cohort (Fig. 5a), respectively. The mean AUCs of time-dependent ROC curves for this ICBcircSig score when predicting the PFS of patients in these three respective cohorts were 0.771, 0.676, and 0.825, respectively, with these values being higher than those for other extant predictors (Fig. 5b). We applied a univariate Cox regression model to assess the relationship between each signature and PFS, and found that 12, 3, and 1 of these 21 signatures were significantly associated with the PFS of patients in Cohort 1, Cohort 2, and our in-house cohort, respectively, with ICBcircSig score achieving the highest HR in all patient cohorts (HR = 4.01, 95% CI: 2.38–6.75, $p < 0.05$ in Cohort 1, HR = 1.35, 95% CI: 1.11–1.66, $p < 0.05$ in Cohort 2, HR = 1.35, 95% CI: 1.22–2.22, $p < 0.05$ in the in-house cohort; Fig. 5c). We further compared the ability of transcriptomic biomarkers to differentiate between ICB responders and non-responders by performing the Wilcoxon Rank Sum test (Fig. 5d, left panel) and univariate logistic regression analyses (Fig. 5d, right panel). The results indicated that ICBcircSig scores were both significantly able to distinguish patients with different responses to ICB treatment and were superior to other predictive tools. Finally, we evaluated the association between each signature and overall survival based on AUC values, and found that ICBcircSig scores also exhibited the highest mean AUC of 0.66 (Supplementary Fig. 5). Taken together, these findings demonstrate that this ICBcircSig score outperforms existing transcriptome-based signatures when predicting responses to ICB treatment.

## The association between ICBcircSig score and cancer hallmarks and immune features

To explore the functional implication of the ICBcircSig score in immunotherapy, we performed an enrichment analysis of 50 hallmark genes[69,70] between high ICBcircSig score and low ICBcircSig score groups based on a gene set enrichment analysis (GSEA) approach. The low ICBcircSig score group was enriched for genes associated with the immune response, including the IFN-α response (normalized enrichment score [NES] = −2.58, $p = 0.0024$), IFN-γ response (NES = −2.72, $p = 0.0027$), inflammatory response (NES = −2.03, $p = 0.0027$), and IL6/JAK/STAT3 signaling pathway (NES = −1.88, $p = 0.0023$) in patient Cohort 1 (Fig. 6a). Consistently, immune response-related cancer hallmark pathways such as the IFN-α response (NES = −2.13, $p = 0.029$) and IL6/JAK/STAT3 signaling pathway (NES = −1.95, $p = 0.023$) were also enriched in the low ICBcircSig score group in Cohort 2 (Supplementary Fig. 6a), suggesting that patients with low ICBcircSig scores may exhibit a more activated immune microenvironment. To further examine whether the ICBcircSig scores were associated with immune infiltration, we conducted GSEA and single sample GSEA (ssGSEA) analyses of the infiltration of 22 immune subpopulations[71] in tumors from patients with low or high ICBcircSig scores. We observed significant differences in immune infiltration when comparing these two ICBcircSig score-based groups (Fig. 6b and Supplementary Fig. 6b) in both tested cohorts. Specifically, 17 and 18 out of the 22 analyzed immune subpopulations exhibited higher levels of predicted infiltration in the low ICBcircSig score group in Cohort 1 and Cohort 2, respectively. These results suggest that patients with low ICBcircSig scores exhibit characteristically high levels of tumor immune infiltration that may explain their higher response rates to immunotherapy. In addition, we assessed the association between ICBcircSig score and cytolytic activity (CYT), a proxy used to reflect the ability of T cells to kill cancer cells[50], and GEP, which corresponds to a T cell-inflamed environment[49], and we observed that the high ICBcircSig score group had a lower CYT score/GEP score than the low ICBcircSig score group in both tested cohorts (Fig. 6c, d and Supplementary Fig. 6c, d), suggesting their less robust cytolytic activity and impaired ability to kill tumor cells.

To validate the functional role of circRNA signatures is based on the circRNA-miRNA-mRNA regulatory axe. We generated the *circTMTC3* overexpression (OE) cell lines and *circFAM117B* OE cell lines in SK-MEL-28 (Fig. 6e, h). According to miranda's prediction27, hsa-miR-142-5p can interact with *circTMTC3* and *circFAM117B*, we observed hsa-miR-142-5p significantly downregulated in OE cell lines of *circTMTC3* or *circFAM117B* (Fig. 6f, i). Then, we found the expression of PD-L1 is significantly upregulated in both OE cell lines by RT-PCR analysis (Fig. 6g, j). These suggest that ICBcircSig signature *circTMTC3* and *circFAM117B* can regulate the hsa-miR-142-5p/PD-L1 pathway in melanoma cell line. To further investigate the functional role of *circTMTC3* and *circFAM117B* in immunosuppression, we performed an in vitro T cell cytotoxicity-mediated tumor killing assay based on SK-

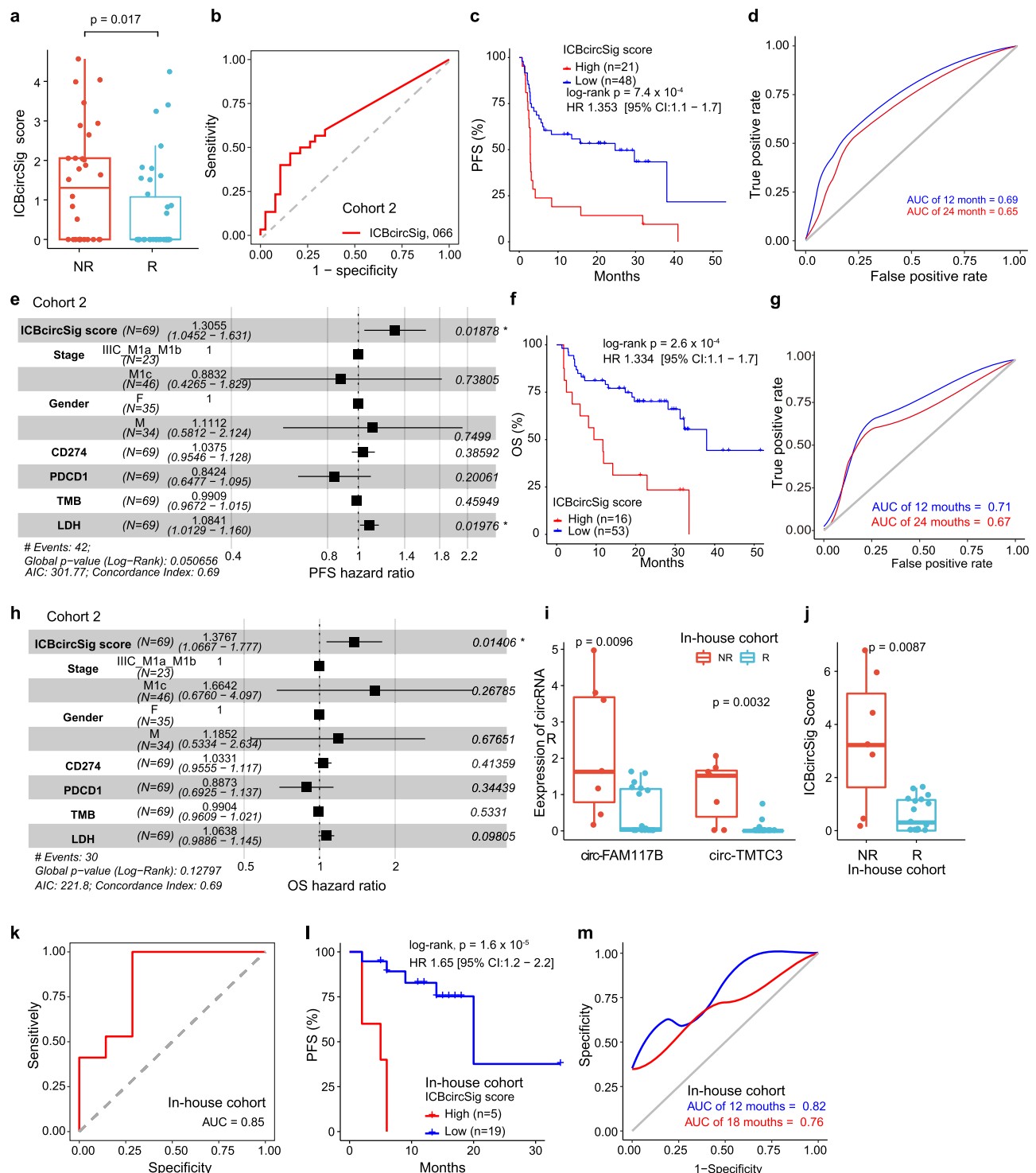

**Fig. 4 | Validation of ICBcircSig score model in two independent cohort.**
**a** Boxplot of ICBcircSig score between response (n = 38) and non-response (n = 30) groups. **b** Receiver Operating Characteristic (ROC) curves quantifying ICBcircSig score prediction AUC. **c** Kaplan–Meier survival curves of PFS between high- and low-ICBcircSig score patients stratified by the optimal cutoff in cohort 2. **d** Time-dependent ROC curve at 12 and 24-months of PFS for the ICBcircSig score. **e** Forest plot for the HRs of multivariate Cox model of the ICBcircSig score and clinicopathological variables. **f** Kaplan–Meier survival curves of OS between high- and low-risk patients stratified by ICBcircSig score using the optimal cutoff in validation data. **g** Time-dependent ROC curve at 12 and 24-months of OS for the ICBcircSig score. **h** Forest plot for the HRs of multivariate Cox model of the ICBcircSig score for OS and clinicopathological variables. **l, j** Boxplot of expression of circTMTC3/circFAM117B (**i**) ICBcircSig score (**j**) distribution between response

(n = 14) and non-response (n = 7) groups for in-house cohort 3. A log-rank test was used in **c**, **f** and **i**. **k** ROC curves quantifying ICBcircSig score response prediction AUC in in-house cohort 3. **l** Kaplan–Meier survival curves of PFS between high- and low-risk patients stratified by ICBcircSig score using the optimal cutoff in in-house cohort 3. **m** Time-dependent ROC curve at 12 and 24-months of PFS for the ICB-circSig score in in-house cohort 3. A two-sided Wilcoxon rank-sum test was used in **a**, **i**, and **j**. The boxes in **a**, **i** and **j** indicate the median ± 1 quartile, with the whiskers extending from the hinge to the smallest or largest value within 1.5× IQR from the box boundaries. Black vertical lines in **e** and **h** indicate the 95% confidence interval (CI). PFS progressive free survival, OS overall survival, ROC receiver operating characteristic curve, HR hazard ratio, AUC Area Under the ROC Curve. Source data are provided as a Source data Fig. 4a–d, f, g, i–l.

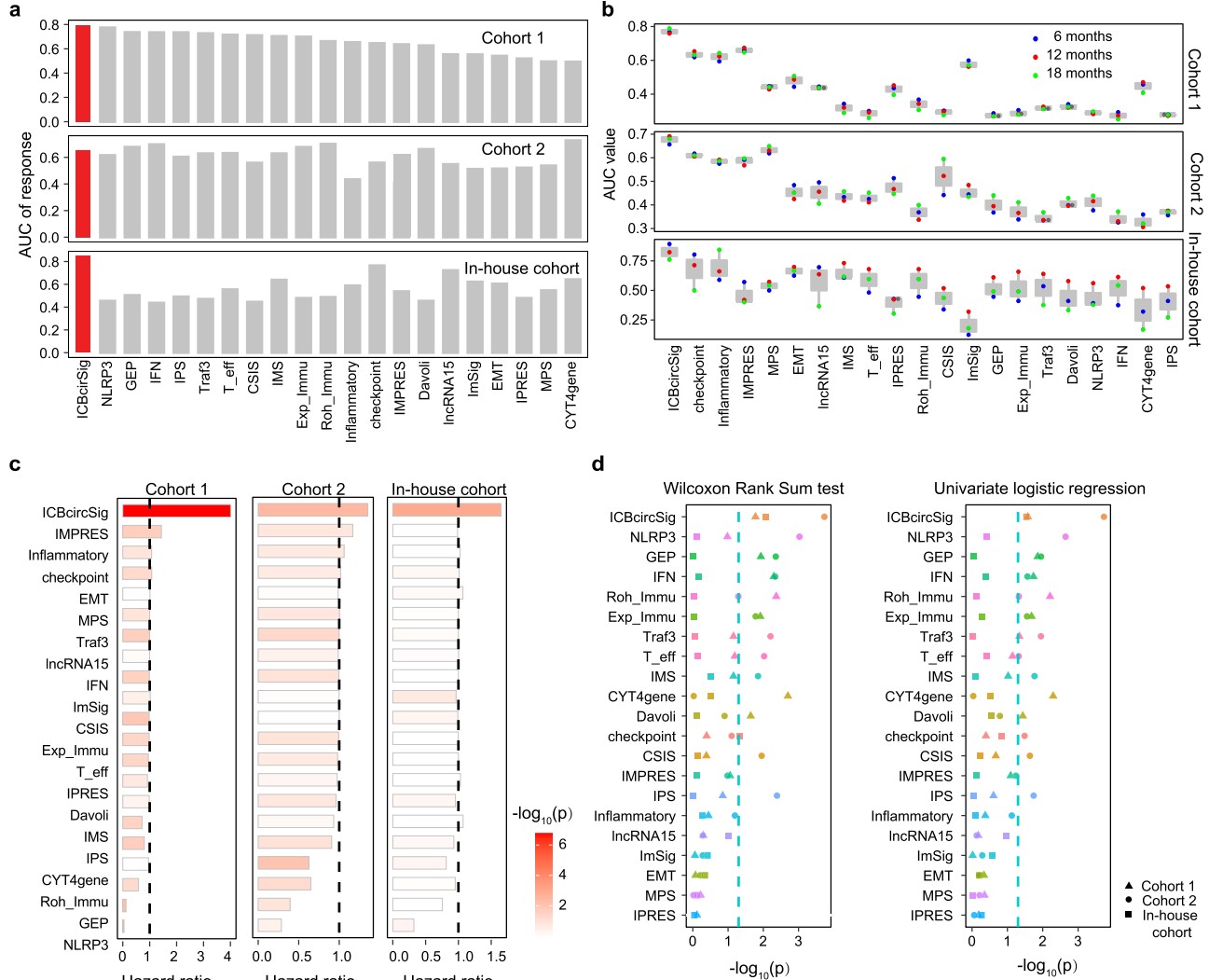

**Fig. 5 | Comparisons of ICBcircSig score model and other transcriptome-based signatures of ICB. a** Performance of ICBcircSig score and other published signatures based on ROC curves quantifying ICBcircSig score response prediction AUC in each cohort. **b** AUC of time-dependent ROC of 6, 12, and 18-months for each cohort. **c** HR for FPS in each cohort. The statistical difference is performed by cox regression model. Red dashed line indicated *p* value 0.05. **d** P value of two-sided Wilcoxon rank-sum test (left panel) and logistic regression test (right panel) between response versus non-response, red dashed line indicated *p* value 0.05. Blue point means 6 months, red point means 12 months and green point means 18-months in **b** (*n* = 3). The boxes in **b** indicate the median ± 1 quartile, with the whiskers extending from the hinge to the smallest or largest value within 1.5× IQR from the box boundaries. ROC receiver operating characteristic curve, HR hazard ratio, AUC Area Under the ROC Curve. Source data are provided as a Source data Fig. 5.

MEL-28 melanoma cells overexpressing *circTMTC3* or *circFAM117B*. The overexpression of *circTMTC3* or *circFAM117B* significantly reduced the CD8[+] T cell cytotoxicity and the ability of these T cells to eliminate tumor cells (Fig. 6k–m). Together, we thus identified a strong correlation between ICBcircSig score and a series of immune signatures that reflect the complex TME, highlighting the prognostic value of this ICBcircSig score and the roles played by *circTMTC3* and *circFAM117B* in immunosuppression.

## Discussion

The roles of circRNAs in cancer are increasingly well understood, but it remains unclear as to whether circRNAs play significant roles in the context of cancer immunotherapy, particularly with respect to whether these circRNAs can serve as biomarkers to predict patient response to ICB treatment. In the present study, we systematically characterized the expression of circRNAs in melanoma patients undergoing ICB treatment in three independent cohorts. We dissected the potential roles of circRNAs in the resistance to cancer immunotherapy through the identification of upregulated cancer signaling pathways, including the Hippo, p53, and mTOR signaling pathways. Importantly, we developed a machine learning-based method to construct an ICBcircSig score to predict the response of ICB-treated melanoma cohorts, which may serve as an independent prognostic factor when adjusting for clinical and molecular features.

Increasing numbers of circRNAs have been reported in the context of tumorigenesis and shown to be related to worse patient prognosis, including circFGFR1[72], circ-CPA4[73], and Circ_0000284[74]. However, the circRNA landscape of ICB-treated cancer patients is not well-characterized, particularly when comparing ICB responders and non-responders. Our study established the circRNA expression landscape from more than 150 ICB-treated patients in two independent cohorts, including pre-treatment and post-treatment tumor tissue samples. We demonstrated that an overall increase in circRNA expression levels in non-responders compared to responders. These upregulated circRNAs can compete with mRNAs for complementary miRNA binding, thereby indirectly regulating mRNA expression. We further characterized circRNA-miRNA-mRNA networks to explore the functional roles of dysregulated circRNAs, and observed that these

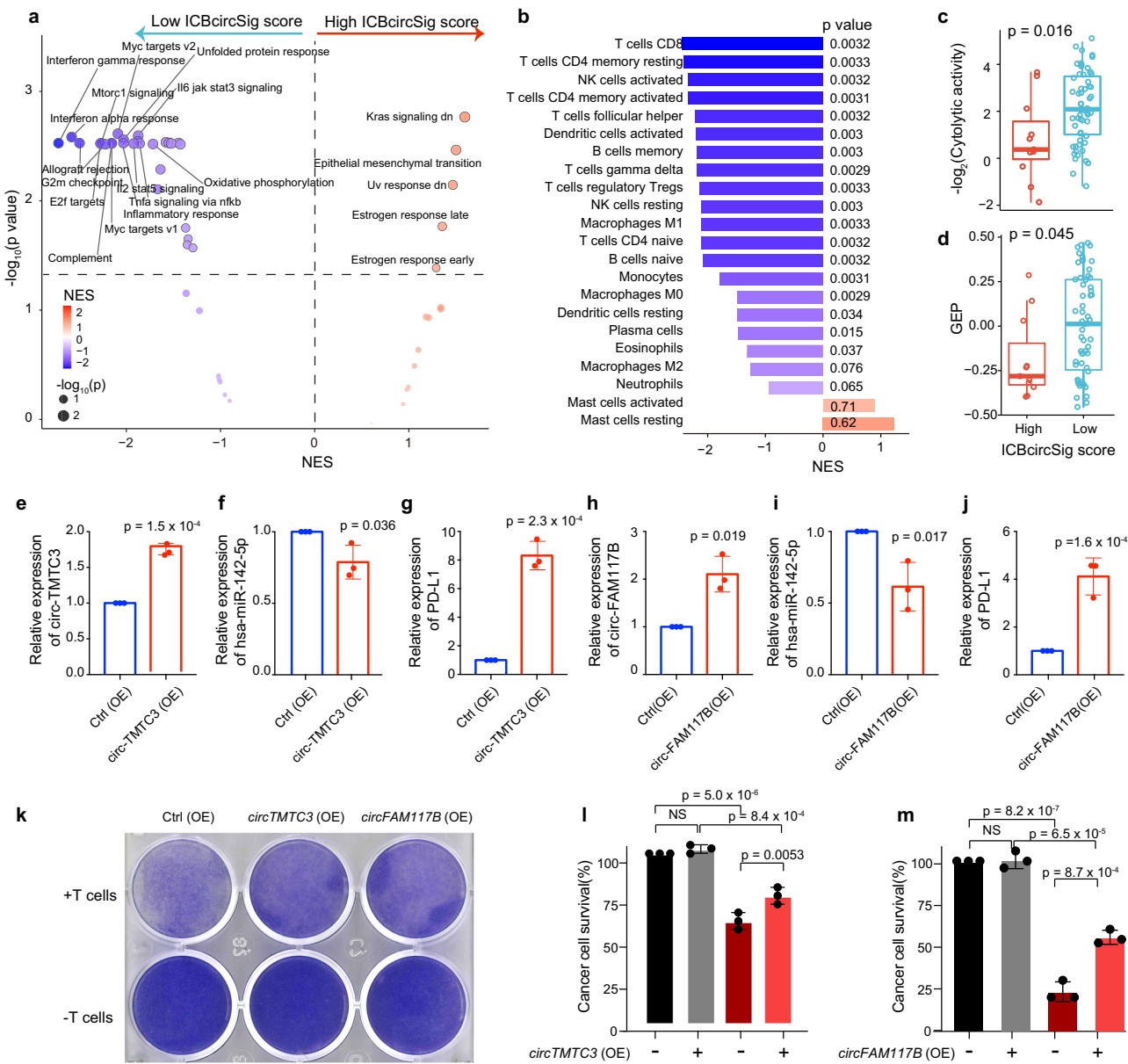

**Fig. 6 | Functional characterization of the ICBcircSig score. a** Volcano plots for the enrichment of hallmarks for cohort 1 samples with high and low ICBcircSig score based on the Normalized Enrichment Score (NES) from the GSEA. **b** The enrichment of immune cell for samples with high and low ICBcircSig score based on the Normalized Enrichment Score (NES) from the GSEA. **c** Cytotoxic T cell score in the high- (n = 11) and low- (n = 59) risk groups stratified by the ICBcircSig score. **d** GEP score in the high- and low-risk groups stratified by the ICBcircSig score. **e**–**g** The expression of *circTMTC3* (**e**), miR-142-5p (**f**), and PD-L1 (**g**) in SK-MEL-28 melanoma cell line with *circTMTC3* overexpression (oe) or control group (n = 3 in each group). **h**–**j** The expression of *circFAM117B* (**h**), miR-142-5p (**i**), and PD-L1 (**j**) in SK-MEL-28 melanoma cell line with *circFAM117B* overexpression (oe) or control group. **k**–**m** SK-MEL-28

melanoma cells with or without overexpression of *circTMTC3* or *circFAM117B* co-cultured with activated T cells for 48 h were subjected to crystal violet staining (**k**). SK-MEL-28-to-T cell ratio, 1:3. **l**, **m** Statistical analysis comparing the ability of T cell killing in (**k**), n = 3 in each group. Two-sided Wilcoxon rank-sum test was used in **c**, **d**. The boxes in **c**, **d** indicate the median ± 1 quartile, with the whiskers extending from the hinge to the smallest or largest value within 1.5× IQR from the box boundaries. The unpaired two-sided Student *T*-test was used in **e**–**j**, one-way ANOVA and Tukey's multiple comparison test was used in **l**, **m**. The error bars in **e**–**j** and **l**, **m** indicate the mean ± s.d. *p < 0.05, **p < 0.01 and ***p < 0.001. Source data are provided as a Source data Fig. 6a–d.

circRNAs were significantly enriched in cancer-related signaling pathways (e.g., Wnt/β-Catenin signaling), and are thus likely to participate in anti-tumor immune and immunotherapy resistance[66]. Therefore, it is vital that further research explore the overall impact of the dysregulation of cancer-related circRNAs on cancer immunotherapy.

CircRNAs have emerged as biomarkers used to predict cancer patient prognosis. However, the value of circRNAs as tools to predict immunotherapeutic efficacy remains to be explored. Our ICBcircSig score model is a highly robust predictor of the efficacy of

immunotherapy in patients with melanoma, irrespective of whether they are undergoing anti-PD-1 treatment alone or combined anti-CTLA-4 and anti-PD-1 treatment. Our ICBcircSig score model performs well with respect to its predictive efficacy in ICB-treated patients as compared to PD-L1 expression status or 20 transcriptome-based features. CircRNAs are noncoding RNAs characterized by a covalently closed circular structure, making them difficult to identify through polyA-enriched RNA-seq analyses[59]. Previous efforts to develop transcriptome-based signatures were based on polyA-enriched RNA-

seq data. Features derived based on polyA-enriched RNA-seq and total RNA-seq data exhibited different levels of predictive power, and it may be of value to integrate these two layers of molecular features into a multi-omics dataset with the goal of improving predictive power in the future. In addition, tumors with a lower ICBcircSig score exhibit higher levels of immune activity, including IFN responses, inflammatory responses, cytolytic activity, NK cells, and CD8+ T cells. Furthermore, we validated ICBcircSig *circTMTC3* and *circFAM117B* promote immune escape through regulating the hsa-miR-142-5p/PD-L1 pathway. These results indicate that the ICBcircSig score is negatively correlated with tumor immune activity, potentially contributing to a less robust response to immunotherapy. In summary, we herein developed a circRNA-based signature to predict immunotherapeutic efficacy, highlighting the relevance of circRNAs in personalized cancer immunotherapy.

## Methods

### Ethics declarations

The study was conducted in accordance with ethical guidelines of U.S. Common Rule. All tissue samples were collected in compliance with the informed consent policy. The study protocol was approved by the Institutional Review Board of Fudan University Shanghai Cancer Center (1906203-3).

### Clinical information and RNA-Seq data of melanoma patients

The raw RNAseq data and clinical information of cohort 1 used in this study were downloaded from the European Nucleotide Archive (ENA) (https://www.ebi.ac.uk/ena) under accession number PRJEB23709. Briefly, melanoma patients were treated with single agent anti-PD-1 (nivolumab or pembrolizumab) or combination anti-PD-1 and anti-CTLA-4 (ipilimumab). Pre-treatment (PRE) indicates biopsies collected prior to immunotherapy, and early during treatment (EDT) means biopsies collected 7–14 days following immunotherapy. At subsequent follow-up, patient response was determined using the Response Evaluation Criteria in Solid Tumors (RECIST) 1.1 criteria, the response group defined as partial/complete response (PR/CR) or stable disease (SD) with PFS > 3 months and non-response group defined as progressive disease (PD) or SD with PFS < = 3 months. A total of 88 samples with RNA-seq data were included in this patient cohort, including 47 patients treated with anti-PD-1 monotherapy (pre-treatment [PRE], $n = 38$; early during therapy [EDT], $n = 9$) and 41 patients treated with combined ipilimumab and anti-PD-1 immunotherapy (patients, $n = 41$; PRE, $n = 32$; EDT, $n = 9$; Supplementary Data 1).

The raw RNAseq data and clinical information of cohort 2 used in this study were downloaded from the database of Genotypes and Phenotypes (dbGaP) (https://www.ncbi.nlm.nih.gov/gap/) under accession number phs000452.v3.p1. Briefly, patients of advanced melanoma were treated with single agent anti-PD-1 (nivolumab or pembrolizumab) with and without previous anti-CTLA-4 treatment. Patient response was assessed using the RECIST 1.1 criteria, which included 30 with progressive disease (PD), 9 with stable disease (SD), 1 with mixed response (MR), 19 with partial response (PR) and 10 with complete response (CR). Biopsies samples collected prior to immunotherapy. A total of 69 samples with RNA-seq data were enrolled in this patient cohort (Supplementary Data 2).

The cohort of 24 patients with melanoma evaluated in this study received anti-PD-1 treatment or combination anti-PD-1 and anti-CTLA-4 were collected between May 2018 and September 2020. They were treated with anti-PD-1 monotherapy ($n = 23$ pembrolizumab 200 mg/cycle every 3 weeks; $n = 1$ ipilimumab 200 mg/cycle every 2 weeks). The median age of the patients was 62.5 years (range, 41 to 85 years), with 9 (37.5 %) male patients and 15 (62.5%) female patients. The clinical information is summarized in Supplementary Table 3.

### Library preparation for total RNA transcriptome sequencing

A total of 1 µg of total RNA per sample was used as input material for lncRNA library preparation. Strand-specific libraries were generated using the NEBNext® UltraTM RNA Library Prep Kit for Illumina® (NEB, USA) according to the manufacturer's recommendations, and index codes were added to the attribute sequence for each sample. o preferentially select cDNA fragments of 150–200 bp in length, the library fragments were purified using the AMPure XP system (Beckman Coulter, Beverly, USA). The second strand was then digested with size-selected adapter-ligated cDNA using USER Enzyme (NEB, USA) for 15 min at 37 °C, followed by 5 min at 95 °C prior to PCR. PCR was then performed with Phusion High-Fidelity DNA Polymerase, Universal PCR Primers, and Index (X) primers. Then, PCR products were purified (AMPure XP system) and library quality was assessed on an Agilent Bioanalyzer 2100 system. Index-encoded samples were clustered on the cBot Cluster Generation System using the TruSeq PE Cluster Kit v3-cBot-HS (Illumina) according to the manufacturer's instructions. Finally, the library preparations were sequenced on Novaseq 6000 platform and 150 bp stand-specific paired-end reads.

### Identification of circRNAs in the ICB samples

Four tools, including CIRI2[56], find_circ[57], CircExplorer2[58], and CircRNA_finder[59] were applied to identify circRNA with default settings. After FastQC (http://www.bioinformatics.babraham.ac. uk/projects/fastqc/) for assessment of the data quality, reads that passed thresholds were aligned to reference genome (GRCh38) using hisat2[75] with the default setting to obtain mapped and unmapped reads in bam files. Unmapped reads were retrieved by samtools from bam files, and unmapped reads in fastq format were done by bedtools bamtofastq. We employed each program to identify circRNAs with default parameters and annotated with gencode_v28. CircRNAs identified by at least two tools with ≥2 back-splice reads were retained for further analysis. The previously identified human tumor circRNAs were downloaded from MiOncoCirc (https://mioncocirc.github.io/)[76].

### Identification of differentially expressed circRNAs between responders and non-responders

To identify differentially expressed circRNAs between responders and non-responders samples in pre-treatment (PRE) and early during treatment (EDT), respectively, a linear mixed-effects model(LME) which allows for nested random effects (each individual sample) and considers for potential confounding factors was utilized and executed by the lme program in the R package[77]. $P < 0.05$ and $|\log_2 (\text{fold change})| \geq 0.5$ was considered as statistical significance.

### Identification of circRNA-miRNA-mRNA regulatory axes

To predict circRNAs-miRNA interactions, we utilize Miranda, which identify potential target sites for miRNAs in genomic sequence[64], to predict target sites of circRNAs. Alignment score and minimum free energy were used to rank the candidate miRNAs for each circRNA. Briefly, sequence information of each circRNA were obtained by bedtools[76] with location of each circRNA and sequence information of each miRNA of human were download from miBase[78]. Then miranda was conducted to predict target sites of miRNAs for each circRNA with following parameters: alignments with energies < = −7 and alignments with scores > = 150). The miRNA−mRNA pairs were obtained from Tarbase[65] with experimentally supported miRNA-mRNA interactions and TargetScan database[79], and kept the shared miRNA−mRNA pairs from two databases. Finally, the circRNA−miRNA−mRNA interaction was filtered as following criteria: (1) we retained 95% of quantile miRNAs expressed in TCGA melanoma in order to filter non-melanoma-related miRNAs; (2) we retained the top 30 miRNAs that interact with circRNA are retained; (3) we kept miRNA that interact with circRNA and mRNA; (4) we retained the circRNA−miRNA−mRNA interaction with

significantly positive Spearman correlation between circRNAs and mRNAs ($Rs > 0.2$ and $p < 0.05$)

## Development of ICBcircSig score model by machine learning

We utilized a machine learning-based algorithm[80] as previously described to construct ICBcircSig. Briefly, (i) we performed univariate survival analysis to identify prognosis relevant circRNAs by assessing the association of progression-free survival (PFS) and the expression of circRNAs; (ii) Based on LASSO Cox regression model, $cv.glmnet$ function in R package glmnet[68] was used. We first set seed 123, and deviance to measure loss to use for cross-validation and 5 folds, to develop the LASSO Cox regression model. Then, we filtered circRNAs with lambda coefficient >0 to retain the optimal combination from circRNAs in (i). The final signature, named "ICBcirSig", include significant circRNAs ($p < 0.05$) by multivariate cox analysis of circRNAs in (ii). (iii) The ICBcircSig score of each sample was built through the following equations based on the expression value and multivariate Cox regression coefficient ($1.001 * circTMTC3 + 1.048 * circFAM117B$).

## Survival analysis

For each circRNA, Kaplan–Meier survival analysis was performed for patients with high and low expression according to the median of the expression level by the R package $survival$. For ICBcirSig score, $Maxstat$ package was used to determine the optimal cutoff point between ICBcirSig score and progression-free survival (PFS) of patient for each cohort. According to the maximum selected log-rank statistics, the patients were divided into the high ICBcirSig score group and low ICBcirSig score group. Subsequently, log-rank test was used to calculate the significance of the differences.

Multivariable cox proportional hazards regression model was used to assess the association of variables with PFS through the $coxph$ function. The R package $survivalROC$ was used to calculate time-dependent receiver-operating characteristic (ROC) curves and area under the ROC curve (AUC) for ICBcirSig score. The R package $pROC$ was used to calculate receiver-operating characteristic (ROC) curves for quantify response for ICBcirSig score.

## Gene set enrichment analysis

Fifty hallmark gene sets were downloaded from The Molecular Signatures Database (MSigDB, http://software.broadinstitute.org/gsea/msigdb/)[69]. A list of 22-tumor-infiltrating immune cell types markers were derived from CIBERSORT. To compare the infiltration alteration of immune cells between ICBcirSig score-high and -low group, we utilize the $clusterProfiler$ package of R to conduct gene set enrichment analysis (GSEA) with marker gene set of each immune cell subpopulation[39]. We calculated signature scores of 22-tumor-infiltrating immune cell types using the GSVA algorithm[81]. Kyoto Encyclopedia of Genes and Genomes (KEGG) and Gene ontology enrichment analysis was carried out for protein-coding genes in the circRNAs-miRNA-mRNA axes by $clusterProfiler$ package[82].

## Cell culture and transfection

The human malignant melanoma cell line SK-MEL-28 cells were cultured in DMEM medium (BI) supplemented with 10% FBS (BI), 100 U of penicillin per ml and 100 mg/ml streptomycin (Gibco) at 37 °C and 5% CO2. A circTMTC3 and circFAM117B overexpression vector was constructed from pHBLV (Hanbio Biotechnology, Wuhan, China). This plasmid contains two repeated sequences named 5'circFrame and 3circFrame, which promote circRNA formation through reverse complementation. Following the manufacturer's instructions, Turbofect (Thermo Fisher) was used for transient transfection of the overexpression vectors.

## RNA isolation and quantitative real-time PCR validation

Total RNA was extracted using TRIzol reagent (Invitrogen, USA) from the tumor biopsy obtained from patients with ICB treatment or in vitro cell line, and then reversed transcribed using the HiFiScript gDNA Removal cDNA Synthesis Kit (YEASEN Biotech, China) according to the manufacturer's instructions. Subsequently, we performed qRT-PCR using SYBR Green assays. GAPDH was used as the reference gene. For cirRNA amplification, RNAs were incubated with RNase R (VWR) 37 °C for 30 min to degrade linear RNAs.RNA was incubated at 70 °C for 10 min to inactivate RNase R and then reverse-transcribed for RT-PCR detection.The levels of miRNAs were measured by qRT-PCR using the miDETECT A TrackTM miRNA qRT-PCR Kit containing a miRNA-specific forward primer (RiboBio, Guangzhou, China) and performed on QuantStudio3 Real-Time PCR System. For the quantitative analysis, relative expression levels were calculated based on CT values (corrected for GAPDH expression) according to the equation: $2^{-\triangle CT}$ [$\triangle CT = CT$ (gene of interest) − CT (GAPDH)]. All qRT-PCR analyses were performed in triplicate. Student's t-tests were applied, and a P-value <0.05 was considered significant.

We used specific primers to PCR $circTMTC3$, F5'-AATACTTCTTACAGGCTACCCATGT-3' and R 5'-AACCACAAAAGAGGCTGTTCC-3'; $circ FAM117B$, F5'-CTTTGCCCAAATATGCAACC-3; and R 5'-CTTTGGAACAGGAGCGAGCA-3'; $PD-L1$, F5'-GATCCAGTCACCTCTGAACATGA-3' and R 5'-TCAGGACTTGATGGTCACTGCT-3'; $GAPDH$, F5'-CAAGGTCATCCATGACAACTTTG-3' and R 5'-GTCCACCACCCTGTTGCTGTAG-3'; U6, F 5'-CTCGCTTCGGCAGCACA-3' and R 5'-AACGCTTCACGAATTTGCGT-3'.

## Sanger sequencing

Total RNA was extracted and treated with RNase R was send out for Sanger sequencing with the primer as 5'-AATACTTCTTACAGGCTACCCATGT-3' for $circTMTC3$ and 5'-CTTTGCCCAAATATGCAACC-3' for $circFAM117B$.

## T cells mediated killing assay

To acquire activated T cells, human peripheral blood mononuclear cells (LTS1077, Yanjin Biological) were cultured in CTS AIIM V serum-free medium (SFM) (A3021002; Gibco) with ImmunoCult Human CD3/CD28/CD2 T cell activator (10970; STEMCELL Technologies) and IL-2 (1000 U mL−1; PeproTech, Rocky Hill, NJ, USA) for 1 week according to the manufacturer's protocol. The experiments were performed with anti-CD3 antibody (100 ng mL−1; 16−0037; eBioscience, Thermo Scientific), interleukin-2 (IL-2), 1000 U mL−1. SK-MEL-28 melanoma cell line from Guangzhou Cellcook Biotech Co., Ltd, which have been authenticated. The cells were seed in the plates overnight and then incubate with activated T cells for 24 h. The ratios between cancer cells and activated cells were 1:3. T cells and cell debris were removed by PBS wash and left cells were quantified by a spectrometer at optical density (OD) 570 nm, followed by crystal violet staining.

## Statistics and reproducibility

All experiments were performed at least three times unless specifically stated. The statistical details and methods is described in the figure legend. Statistical analysis for qRT-PCR and T cells mediated killing assay was performed using GraphPad Prism 9.0. Data are expressed as mean ± SD (standard deviation). Unpaired two-tailed Student's $t$ test was used to analyze the differences between two groups. Comparisons among multiple groups were analyzed with one-way analysis of variance. Differences between high and low ICBcirSig score groups were assessed using Wilcoxon rank-sum test and Univariate logistic regression, were performed in R version 3.6. $P < 0.05$ was considered statistically significant.

**Reporting summary**

Further information on research design is available in the Nature Portfolio Reporting Summary linked to this article.

## Data availability

The raw data of bulk RNA-seq generated in this study have been deposited in the Genome Sequence Archive (GSA) under accession code HRA003368. The raw data of bulk RNA-seq data are available under restricted access for data privacy laws related to patient consent for data sharing, access can be obtained by requesting and following the guidelines for GSA for non-commercial use at https://ngdc.cncb.ac.cn/gsa-human/request/HRA003368. The raw data of cohort 1 and 2 were downloaded from the European Nucleotide Archive (ENA, PRJEB23709) and the database of Genotypes and Phenotypes (dbGaP, phs000452.v3.p1). The processed gene expression data of cohort 1, cohort 2, and in-house cohort were submitted as Supplementary Data 3, 4, and 7. The data information (e.g., sample size, overall survival times, progressive free survival time) of cohort 1, cohort 2, and in-house cohort were summarized in Supplementary Data 1, 2, and Supplementary Table 2. The sequence information of miRNAs of human were download from miBase (https://www.mirbase.org/)[78]. The miRNA–mRNA pairs were obtained from Tarbase (http://www.microrna.gr/tarbase)[65] and TargetScan (https://www.targetscan.org/)[79] database. Source data are provided with this paper.

## Code availability

Codes were implemented in R 3.6.0 and are deposited in https://github.com/Yelab2020/ICBcircSig and achieved at https://doi.org/10.5281/zenodo.7771722[83].

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

## Acknowledgements

This work was supported by grants from the National key research and development program (2022YFC2504700 to Y.Y. and X.C.), the National Natural Science Foundation of China (81971487 to Y.Y., 82203189 to Y.Dong, and 81902149 to Q.G.), the Natural Science Foundation of Shanghai (20ZR1472900 to Y.Y.), Shanghai Jiao Tong University 2030 Initiative (WH510363001-4 to Y.Y.), Shanghai Science and Technology Commission (20JC1410100 to Y.Y.), the National Key Research and Development Program of China (2019YFA0111600 and 2019YFE0120800 to H.L.), Key Program of National Natural Science Foundation of China (U22A20329 to H.L.), the Natural Science Foundation of China for outstanding Young Scholars (No.82022060 to H.L.), The science and technology innovation Program of Hunan Province (2022RC3004 to H.L.), Science Found for Creative Research Groups of the National Natural Science Foundation of China (82221002 to H.L.). Central South University Research Program of Advanced Interdisciplinary Studies (2023QYJC004 to H.L.) Natural Science Foundation of Hunan Province (2020JJ5892 to Q.G.). Key Program of National Natural Science Foundation of China (82130090 and 81830096 to X.C.).

## Author contributions

Y.Y. conceived and supervised the project. Y.Y., L.H., and Y.Dong designed and performed the research. Y.Y., Z.Z., Y.Dong, and Y.Du performed data analysis. G.W. performed the experiment. Y.C., Q.G., G.Z., X.C., and H.L. obtained patient data with immunotherapy. Q.G. performed experiment. Y.Y., S.L., Y.Du., H.L., X.C., L.H., and Y.Y. interpreted the results. Y.Dong., Q.G., L.H., and Y.Y. wrote the manuscript.

## Competing interests

The authors declare no competing interests.
