## [Peer Review File · Nature Communications]

Identification of CircRNA signature associated with tumor immune infiltration to predict therapeutic efficacy of immunotherapyREVIEWER COMMENTS

Reviewer #1 (Remarks to the Author): expertise in immune checkpoint therapy in melanoma

This report identified circulating RNA signature that was predictive of PFS in two published cohorts of melanoma patients treated with immune checkpoint inhibitors.

Generally the manuscript needs a thorough review to improve consistency and grammar - I have highlighted several examples below. There is also insufficient information and rationale provided. In terms of the circRNA signature - there are limited details. I could not find specific details of how many or which circRNAs were included or the final expression equation (25?) - I assume it isn't based on the circTMTTC3 and circ-FAM117B circRNAs only?

The AUC of the score was also below 0.7 in cohort 2, and this highlights the need for further confirmation. There are many other available RNA sequence data from ICI treated melanoma cohorts - see Auslander et al for a starting list and EGAD000001005738

Importantly, the additional cohort used was all acral lentiginous melanoma - a distinct cutaneous melanoma with poorer response to ICI. The stats are borderline for this validation data shown in Figure 3E and 3G

There are also important conclusions that required clarification. The IMPRES signature is an example of this - in 9 different cohort the AUC of this signature ranged from 0.73-0.96, and yet in the two cohorts in this report the IMPRES AUC was below 0.6.

Line 51: needs revision...most frequent histological subtype' seems incomplete

Line 64 - some improvement necessary - 'circRNAs are participants' consider rewording

Line 65-69 needs rewrite as not clear

Line 70: some clarification on the abnormal circRNAs made by tumor cells is required? i.e how are they abnormal?

Line 76 when mentioning biomarkers - key details should be included for instance such as tumour PD-L1 expression, CD8 T cell infiltration etc. the mention of 'gene expression signatures' is meaningless

Line 93: benefit and not-benefit - for improved clarity define these categories here

I assume that Figure 1E is based on the conditions of circRNAs identified by at least two tools with back-splicing reads ≥ 2 - this should be stated in the legend. I am also curious with this filtering is there now a correlation between unmapped reads and number of circRNAs?

Line 114: why was the 20% threshold selected - was this based on downstream prediction analysis - i.e 20% yielded a predictive signature?

Line 128: The MiOncoCirc database conclusion needs clarifying - it seems that the only conclusion that can be made is that the circRNAs have also been identified in various tumours - not that they are derived from tumour cells - which is implied here

There is insufficient reasoning provided for selecting the shared PRE-EDT 81 circRNAs. The authors have assumed that circRNA expression at PRE must be elevated in order to be elevated at EDT, this may be reasonable, but no details have been provided and it appears as though a post hoc strategy was applied.

There are inconsistencies between the S1 and S2 Tables. Patient No are not included in Table S1, treatment columns are not consistent, age is only included in Table S1, biopsy site is only in S2. Add footnote to Tables if the data are not available. Acral is misspelt in S5

Line 135 when identifying circRNA in PRE benefit vs non-benefit - why is FDR-adjusted P value applied?

The 227 up regulated circRNAs were defined from analysis of cohort 1 and it is not surprising that markers separating PD from SD/PR/CR patients would show PFS differences? What about overall survival?

Was treatment included in multivariate analysis, tumour burden, LDH?

Additional details on the performance of these signatures for responders vs non-responders should

be defined in supplementary data - i.e I would like to see performance AUC or HR for PD/SD vs PR/CR for all signatures

Line 259 Many of the signatures tested are related to active immune environment - but the performance is not equivalent. This suggests that there is more to prediction than immune activity – as suggested here

Reviewer #2 (Remarks to the Author): expertise in circRNA analysis from RNA-seq

In the manuscript entitled "Identification of CircRNA Signature Associated with Tumor Immune Infiltration to Predict Therapeutic Efficacy of Immunotherapy", Dong et al investigated the expression landscape of circRNA in 157 immune checkpoint blockades (ICB) treated patients with advanced melanoma. The authors demonstrated the up-regulated circRNA expression between non-benefit patients compared to benefit patients, and these circRNAs could regulate tumor-related pathways through the circRNA-miRNA-mRNA regulatory axes. Besides, the authors also constructed a novel ICBcircSig score for predicting the efficacy of ICB treatment, which outperformed 20 state-of-art signatures and could serve as a useful prognostic factor.

The manuscript is clearly written and the conclusions are well supported by their experiments and data analyses. This study provides novel insights into the potential application of circRNA signatures in ICB therapy, which will be of great interest to the community. I'd like to recommend it to be published if the authors could address the following concerns.

Here are the specific comments:

1. In lines 126-127, the authors compared the detected circRNA and MiOncoCirc which are composed of clinical cancer samples and cell lines. What's the overlap of these circRNAs in other databases that mainly include normal tissue samples (e.g. circAtlas)? What's the proportion of circRNAs that are conservatively expressed in both normal and tumor samples?
2. In lines 161-163, the authors predicted the potential targets of the circRNA-miRNA-mRNA regulatory axis and performed pathway enrichment analysis. Are these mRNAs also up-regulated expressed between non-benefit and benefit patients? The authors could also perform differential expression analysis of linear genes then compare the pathway enrichment results using up-regulated genes subsequently.
3. The authors identified circ-TMTC3 and circ-FAM117B as promising biomarkers for prognosis prediction. What's the presumable function of these two circRNAs in regulating the response to ICB treatment. Is there any evidence that these circRNAs could also bind cancer-related miRNAs?
4. In Fig.3J, Fig.4B, and Fig.4E, the authors used the AUC method to evaluate the performance of the ICBcircSig score. However, the details of calculating the AUC lines are absent in the Methods section. Considering that all ROC curves are smooth curves rather than broken lines, were these results calculated from repeated cross-validation? The authors should provide more details in the Methods section.

Minor comments:

1. "Find_circ" should be written as "find_circ", which is more commonly used.
2. The confidence interval could be added to the ROC curve if repeated measures are involved.
3. There are many grammar mistakes and misused words. For example, "were download from" (line 315, line326), "ICBcircSig score outperforms and other" (line 229), "a novel biomarkers" (line 300), "the prediction of efficacy" (line 304). The language should be edited by native speakers.
4. The authors should provide more technical details on how to identify DE circRNAs and construct

the ICBcircSig models, including softwares, parameters, etc. It seems that the authors used a simple statistical model to detect differentially expressed circRNAs, which may generate false positives. A more sophisticated approach CIRIquant should be used or benchmarked here.

5. Several bioinformatic tools and databases used in this study should be cited. For example, the dataset of PRJEB23709, FastQC, samtools, CIRI2(PMID: 28334140, PMID: 25583365), Kaplan-Meier survival analysis.

Reviewer #3 (Remarks to the Author): expertise in bioinformatics analysis of transcriptomic data

In this work, authors have identified some key immune-related circRNA in melanoma. Taking advantage of the patients with immune checkpoint blockades treatment, a circRNA-miRNA-mRNA regulatory axes was developed. However, it has not been discussed in depth. Besides, authors have constructed and validated a ICBcircSig signature to distinguish ICB non-benefit patients and benefit patients. There were some issue need to be solved.

Major point:

1. In the method 'Clinical Information and RNA-Seq data of melanoma patients', author gives the clinical information of melanoma patients prior to immunotherapy (PRE) and patients early during treatment. However, patient's response was determined for the PRE. Why the patients without treatment can be classified into responder and non-responder? It's better to explain it more clearly.
2. How author predict miRNA-circRNA interaction by Miranda? Please give a detailed description, especially for the confidence of predicted targets. Moreover, authors can validate the interactions in transcription level.
3. Functional analysis based on circRNA-miRNA-mRNA regulatory axes needs more explanations for immune correlations. And it is better to prove the relevance of immune regulation through the experiment.
4. Authors should explain the process of GSEA for accessing the function enrichment in patients with high ICBcircSig score and low ICBcircSig score as detailed as possible.
5. In this work, author identified the immune-related circRNAs by a series of computational methods. However, the mechanism of these genes mediating immune regulation in ICB treatment has not been discussed. Authors should provide more evidence to support the result, especially experimental verification.

Minor point:

1. There is an incorrect uppercase format in the title of method 'Clinical Information and RNA-Seq data of melanoma patients'.
2. It seems that there is a spelling mistake in line 366 of method 'Identification of circRNA-miRNA-mRNA regulatory axes'.
3. Please give the detailed parameters during developing the LASSO Cox regression model.

RESPONSE TO REVIEWERS' COMMENTS:

Reviewer #1 (Remarks to the Author): expertise in immune checkpoint therapy in melanoma

This report identified circulating RNA signature that was predictive of PFS in two published cohorts of melanoma patients treated with immune checkpoint inhibitors.

Generally the manuscript needs a thorough review to improve consistency and grammar - I have highlighted several examples below. There is also insufficient information and rationale provided. In terms of the circRNA signature – there are limited details. I could not find specific details of how many or which circRNAs were included or the final expression equation (25?)– I assume it isn't based on the circTMTC3 and circ-FAM117B circRNAs only?

Response: We thank the reviewer's suggestions. We re-wrote in order to keep the consistency in the whole manuscript and provide sufficient information and rationale in the revised manuscript. In addition, we invited the professional English-language editing service to improve the grammar and sentence structure.

The final circRNA signature consisting of two circRNA (circTMTC3 and circ-FAM117B) and the detailed process of development of the signature is described in Method section "***Development of ICBCircSig signature by machine learning***" (page 18, line 466-476). The equation of ICBCircSig score is $(1.001 * \text{circ-TMTC3} + 1.048 * \text{circ-FAM117B})$, which is also clarified in the Method section.

The AUC of the score was also below 0.7 in cohort 2, and this highlights the need for further confirmation. There are many other available RNA sequence data from ICI treated melanoma cohorts - see Auslander et al for a starting list and EGAD000001005738

Response: We thank the reviewer's great suggestion. We carefully examined the studies of Auslander *et al.*¹ and found anti-PD1/PD-L1 RNA-seq related works (GSE115821¹, Chen², GSE78220³, GSE93157⁴, GSE91061⁵, TCGA⁶, but we did not find the dataset with accession ID as EGAD000001005738). These datasets were performed by polyA enriched RNA-seq. The circRNA is noncoding RNA characterized by a covalently closed circular structure, which makes it difficult to be identified by polyA enriched RNA-seq⁷.

In the previous **Fig. 4f**, although the AUC in cohort 2 is below 0.7, but it is still higher than any other methods regarding the prediction for progression-free survival (PFS) (**Fig. 5b**), and comparable to other methods regarding the prediction for response (**Fig 5a**).

To further confirm, we collected an additional in-house cohort of 24 melanoma patients with anti-PD-1 treatment (see Methods, **Table S6**), including 18 responders and 7 non-responders. Consistently, two circRNAs in our model are highly expressed in non-responders (**Additional Figure [Fig. A]1a**), and our circRNA model score in responders is significantly higher than non-responders (Wilcoxon test, $p = 0.0051$; **Fig. A1b**). The receiver-operating characteristic (ROC) quantified ICBCircSig score prediction and found that the AUC achieved 0.85 in our in-house cohort, suggesting the robust prediction performance of ICBCircSig score (**Fig. A1c**). In addition, patients with a higher score had worse progression free survival (PFS) (log-rank test, $p = 0.032$;

Fig. A1d). The AUCs of the time-dependent ROC curves of the ICBcircSig score were 0.82 and 0.76 for PFS at 12 month and 24 months (**Fig. A1e**). We added **Fig. A1** as **Fig. 4i-m** and the results of in-house cohort in the revised manuscript (page 11, line 268-276).

Fig. A1. Validation of ICBcircSig score model in in-house cohort (a-b) Boxplot of circ-FAM117B, circ-TMTC3 (**a**), and ICBcircSig score (**b**) distribution between responders (R) and non-responders (NR). (**c**) ROC curves quantifying ICBcircSig prediction AUC in in-house melanoma cohort. (**d**) Kaplan–Meier survival curves for in-house melanoma cohort with anti-PD-1 treatment. (**e**) Time-dependent ROC curve at 12 and 18-months of PFS for the ICBcircSig score. The boxes in **a** and **b** indicate the median \pm 1 quartile, with the whiskers extending from the hinge to the smallest or largest value within $1.5 \times$ IQR from the box boundaries. A two-sided Wilcoxon rank-sum test was used in **a** and **b**; a two-sided log-rank test was used in **d**. PFS, progressive free survival; R, responder; NR, non-responder; ROC, receiver operating characteristic curve; HR, hazard ratio; AUC, Area Under the ROC Curve.

Importantly, the additional cohort used was all acral lentiginous melanoma – a distinct cutaneous melanoma with poorer response to ICI. The stats are borderline for this validation data shown in Figure 3E and 3G

Response: Indeed, patients with acral lentiginous melanoma had worse response to ICI. The relatively small sample size (responders, $n = 11$; non-responders, $n = 4$) maybe the reason for the

borderline of statistical analysis in Figure 3E and 3G. To address this concern, we further collected total RNA-seq for an additional in-house cohort of 24 melanoma patients with anti-PD-1 treatment, including acral melanoma (n = 16), cutaneous melanoma (n = 2), mucosal melanoma (n = 2), nodular melanoma (n = 3) and unknown type (n = 1), we found circ-FAM117B and circ-TMTC3 significantly upregulated in non-responders (**Fig. A1a**; circ-FAM117B: p = 0.0096, circ-TMTC3: p = 0.0032). The ICBcircSig score model can accurately predict the response of immunotherapy, regardless of the subtype of melanoma. Further analysis with large-scale sample size is necessary. We added the result of additional in-house cohort in the revised manuscript.

There are also important conclusions that required clarification. The IMPRES signature is an example of this – in 9 different cohort the AUC of this signature ranged from 0.73-0.96, and yet in the two cohorts in this report the IMPRES AUC was below 0.6.

Response: We thank the reviewer’s comment. In our previous manuscript, we calculated IMPRES from the code provided in GitHub (https://github.com/olapuentesantana/easier_manuscript/blob/main/R/compute.IMPRES.R). The code in GitHub calculated IMPRES scores by applying ratio of 15 checkpoint gene pairs. We carefully checked the IMPRES signature of the original paper. IMPRES scores were calculated by applying logical comparisons of 15 checkpoint gene pairs. We recalculated the IMPRES scores and calculated AUC values according to the method described in Auslander et al., and the results are consistent with the values of the original paper for 9 different cohorts. To be noticed, they used IMPRES signature to predict response, not PFS as in our study. We then used IMPRES to predict response in our cohorts, consistently obtained relatively low score 0.64, 0.62, and 0.54 in cohort 1, cohort 2, and in-house cohort, respectively.

The potential reason for IMPRES achieved the low AUC may be that the development of IMPRES is based on the polyA-enriched RNA-seq data, while we used total RNA-seq to quantify circRNAs, but this need further investigation, which might be beyond the scope of our study. We discussed this in the Discussion section in the revised manuscript (page 14-15, line 375-381).

Line 51: needs revision...most frequent histological subtype' seems incomplete

Response: We revised the sentence “Melanoma is the most common histological subtype of skin cancer” in the revised manuscript (page 4, line 54).

Line 64 - some improvement necessary – ‘circRNAs are participants' consider rewording

Response: We corrected “circRNAs are participants” to “circRNAs are involved in”.

Line 65-69 needs rewrite as not clear

Response: We rewrote the sentences as follows: “For example, the circRNA hsa_circ_0020397 can bind and inhibit the expression of miR-138, which targets PD-L1 to inhibit its expression. Therefore, the overexpression of hsa_circ_0020397 promotes the upregulation of PD-L1, leading to immune escape in colorectal cancer⁸. CircRNAs may also interact with proteins. For example, circFoxo3 can regulate p53-influenced immune responses by inducing the ubiquitination-dependent degradation of p53 through binding to MDM2^{9,10}.” in the revised manuscript (page 4, line 68-73).

Line 70: some clarification on the abnormal circRNAs made by tumor cells is required? i.e how are they abnormal?

Response: We re-wrote the sentences as follows: “Tumor cells may produce abnormal circRNAs caused by genetic mutations¹¹, chromosomal translocation¹², TGF- β signaling regulation¹³, and other aberrant events¹⁴. For example, PML/RAR α chromosomal translocations lead to the generation of fusion circRNAs (F-circRNAs) in acute promyelocytic leukemia, which promote cellular transformation, cell viability, and resistance to therapy¹⁵” in the revised manuscript (page 4, line 73-77).

Line 76 when mentioning biomarkers - key details should be included for instance such as tumour PD-L1 expression, CD8 T cell infiltration etc. the mention of 'gene expression signatures' is meaningless.

Response: We strongly agree with the reviewer that PD-L1 expression and CD8 T cell infiltration are important biomarkers for predicting immunotherapy efficacy¹⁶⁻¹⁸, and we described in the revised manuscript as follows “To date, several biomarkers associated with responses to ICB in melanoma patient have been identified. Tumor mutational burden (TMB) (≥ 10 mutations/megabase) has been approved as an ICB therapeutic biomarker for the treatment of unresectable or metastatic solid tumors with pembrolizumab^{19,20}. PD-L1 expression, as determined by immunohistochemistry (IHC), is also used clinically as a companion diagnostic biomarker²¹⁻²⁴. CD8+ T cells, which are critical for tumor cell recognition and killing²⁵, have been identified as a positive biomarker to predict ICB responses in multiple cancer types²⁶” (page 5, line 83-90).

In this study, we focused on the circRNA, one type of transcriptome-based signatures. We revised the statement as “transcriptome-based signatures”^{1,27,28} in revised manuscript (page 5, line 90-94).

Line 93: benefit and not-benefit – for improved clarity define these categories here

Response: The benefit group (responder) was defined as partial/complete response (PR/CR) or stable disease (SD) with PFS > 3 months, and non-benefit group (non-responder) was defined as progressive disease (PD) or SD with PFS \leq 3 months. We clarified these categories in the revised manuscript (page 7, line 159-161).

I assume that Figure 1E is based on the conditions of circRNAs identified by at least two tools

with back-splicing reads ≥ 2 - this should be stated in the legend. I am also curious with this filtering is there now a correlation between unmapped reads and number of circRNAs?

Response: We added “circRNAs identified by at least two tools with back-splicing reads ≥ 2 ” in the legend of **Fig. 1e**.

We performed a correlation analysis between unmapped reads and the number of circRNAs in **Fig. 1e**. We found no significant correlation (**Fig. A2**).

Fig. A2. The correlation between unmapped reads and number of circRNAs. Spearman correlation between the number of unmapped reads and the number of circRNAs identified by four circRNA-detection tools.

Line 114: why was the 20% threshold selected - was this based on downstream prediction analysis -i.e 20% yielded a predictive signature?

Response: We have combined four well-established circRNA detection tools to quantify circRNAs for the most reliable precision and sensitivity with balanced performance¹³. We aim to identify a reasonable number of differentially expressed circRNAs between responders and non-responders, we tested several cutoffs range from 10%-80%, and obtained the same number of differentially expressed circRNAs between responders and non-responders when using cutoff as 10% and 20%, while this number will decrease sharply if we set the cutoff as 30%. We therefore set 20% as the cutoff.

Line 128: The MiOncoCirc database conclusion needs clarifying - it seems that the only conclusion that can be made is that the circRNAs have also been identified in various tumors - not that they are derived from tumor cells - which is implied here

Response: We agree with the reviewer. To avoid the confusion of this conclusion, we revised this sentence as “suggesting circRNAs in these two cohorts have also been identified in various tumor tissues” in the revised manuscript.

There is insufficient reasoning provided for selecting the shared PRE-EDT 81 circRNAs. The authors have assumed that circRNA expression at PRE must be elevated in order to be elevated at EDT, this may be reasonable, but no details have been provided and it appears as though a post hoc strategy was applied.

Response: We thank the reviewer’s suggestion. Indeed, we have assumed that circRNA expression at PRE may be elevated in order to be elevated at EDT when we constructed the ICBcircSig score based on the circRNA expression at PRE samples. We added the detail description in the revised manuscript (page 8, line 173-174)

There are inconsistencies between the S1 and S2 Tables. Patient No are not included in Table S1, treatment columns are not consistent, age is only included in Table S1, biopsy site is only in S2. Add footnote to Tables if the data are not available. Acral is misspelt in S5.

Response: We corrected the content and make the consistence of text in **Supplementary Table 1 and 2**.

Line 135 when identifying circRNA in PRE benefit vs non-benefit - why is FDR-adjusted P value applied?

Response: We did not perform FDR-adjustment in our original manuscript.

The 227 up regulated circRNAs where defined from analysis of cohort 1 and it is not surprising that markers separating PD from SD/PR/CR patients would show PFS differences? What about overall survival?

Response: We performed the association analysis between ICBcircSig score and overall survival and found our circRNA signature could also stratify the patient survival (hazard ratio [HR] = 1.637, 95% confidence interval [95%CI] 0.77–3.5; $p = 0.028$; **Fig. A3**). We added the **Fig. A3** as **Fig. 3n** in the revised manuscript (page 10, line 243-244).

Fig. A3. Kaplan–Meier survival curves of patients with high versus low ICBcircSig score. The P value is computed via a two-sided log-rank test.

Was treatment included in multivariate analysis, tumour burden, LDH?

Response: We assessed whether our ICBcircSig score is also an independent predictor considering mutation burden and LDH by performing multivariate Cox regression analysis in cohort 2. We found that our circRNA signature was also an independent predictor significantly associated with worse PFS (hazard ratio [HR] = 1.30, 95% confidence interval [95%CI] 1.04–1.63; p = 0.018; **Fig. A4a**) and worse OS (hazard ratio [HR] = 1.38, 95% confidence interval [95%CI] 1.06–1.77; p = 0.014; **Fig. A4b**). We added the **Fig. A4** as **Fig. 4e and h** and the result in revised manuscript (page 11, line 258-263, and 266-267).

Fig. A4 Multi-Cox model of ICBcircSig score. Forest plot showing the PFS (a) and OS (b) HRs of multivariate Cox model of the ICBcircSig score, clinicopathological variables (tumor stage, gender), and ICB related features (expression of CD274 and PDCD1, TMB, Tx_Start_LDH). PFS, progressive free survival; OS, overall survival; HR, hazard ratio; TMB, tumor mutation burden; LDH, lactate dehydrogenase.

Additional details on the performance of these signatures for responders vs non-responders should be defined in supplementary data - i.e I would like to see performance AUC or HR for PD/SD vs PR/CR for all signatures

Response: We thank the reviewer for this comment. We used score of all signatures and the clinical response data to generate the ROC classification curves estimating the prediction performance to achieve AUCs in cohort 1, cohort 2, and our in-house cohort (**Fig. A5**). We have added **Fig. A5** as revised **Fig. 5a** in the revised manuscript (page 11, line 281-283).

Fig. A9 Response predictive performance of ICBcircSig and 20 transcriptome-based features. ROC curves quantifying ICBcircSig and 20 transcriptome-based features prediction AUC across cohort 1, cohort 2, and our in-house cohort.

Line 259 Many of the signatures tested are related to active immune environment - but the performance is not equivalent. This suggests that there is more to prediction than immune activity – as suggested here

Response: We agree with the reviewer. We discussed this in the Discussion section (page 14-15, line 369-384).

Reviewer #2 (Remarks to the Author): expertise in circRNA analysis from RNA-seq

In the manuscript entitled “Identification of CircRNA Signature Associated with Tumor Immune Infiltration to Predict Therapeutic Efficacy of Immunotherapy”, Dong et al investigated the expression landscape of circRNA in 157 immune checkpoint blockades (ICB) treated patients with advanced melanoma. The authors demonstrated the up-regulated circRNA expression between non-benefit patients compared to benefit patients, and these circRNAs could regulate tumor-related pathways through the circRNA-miRNA-mRNA regulatory axes. Besides, the authors also constructed a novel ICBcircSig score for predicting the efficacy of ICB treatment, which outperformed 20 state-of-art signatures and could serve as a useful prognostic factor.

The manuscript is clearly written and the conclusions are well supported by their experiments and data analyses. This study provides novel insights into the potential application of circRNA signatures in ICB therapy, which will be of great interest to the community. I'd like to recommend it to be published if the authors could address the following concerns.

Response: We thank the reviewer for overall positive comments that highlight the significance of our study.

Here are the specific comments:

1. In lines 126-127, the authors compared the detected circRNA and MiOncoCirc which are composed of clinical cancer samples and cell lines. What's the overlap of these circRNAs in other databases that mainly include normal tissue samples (e.g. circAtlas)? What's the proportion of circRNAs that are conservatively expressed in both normal and tumor samples?

Response: Thanks for the reviewer's valuable comment. We performed overlap analyses among the two cohorts in this study, circAtlas²⁹, and the MiOncoCirc³⁹ database (**Fig. A6**). A significant overlap of detectable circRNAs was observed between Cohort 1 and Cohort 2, with 90.4% (3,293/3,644) of the circRNAs in Cohort 2 having been detected in Cohort 1 (Fisher test, $p < 2.2e-16$; Fig. 1i). We further found that 96.6% of the circRNAs (5,168/5,350) in Cohort 1 and 97.3% (3,544/3,654) in Cohort 2 were identified in MiOncoCirc database, suggesting circRNAs in these two cohorts have also been identified in various tumor tissues. 98.5% of the circRNAs (5,271/5,350) in Cohort 1 and 98.8% (3,612/3,654) in Cohort 2 were identified in the circAtlas database. Furthermore, 95.4% of circRNAs (5,106 /5,350) in Cohort 1 and 96% (3,508/3,654) in Cohort 2 were identified in both the circAtlas and MiOncoCirc databases, suggesting the conserved expression of these circRNAs in both normal and tumor samples (page 7, line 144-154).

Fig. A6. The overlap of identified circRNAs between cohort 1 or cohort 2 and public circRNA database circAtlas and MiOncoCirc.

2. In lines 161-163, the authors predicted the potential targets of the circRNA-miRNA-mRNA regulatory axis and performed pathway enrichment analysis. Are these mRNAs also up-regulated expressed between non-benefit and benefit patients? The authors could also perform differential expression analysis of linear genes then compare the pathway enrichment results using up-regulated genes subsequently.

Response: We calculated the differentially expressed mRNAs between non-benefit and benefit patients by Wilcoxon Rank Sum test, resulting 151 up-regulated and 299 down-regulated mRNAs ($|\log_2(\text{fold change})| > 0.5$ & $p < 0.05$) and found that up-regulated mRNAs are significantly overlapped with our circRNA-related mRNA based on the network of circRNA-miRNA-mRNA regulatory axis ($p < 2.2e-16$ using Fisher's exact test; **Fig. A7**). Since there are few up-regulated genes in benefit groups, we did not further perform pathway enrichment analysis.

Fig. A7. Overlap of circRNAs related genes and upregulated genes in non-benefit group. Venn diagram between 151 up-regulated genes in benefit group (red) and circRNA-related mRNA based on the network of circRNA-miRNA-mRNA regulatory axis (orange). P-value calculated by Fisher's exact test.

3. The authors identified circ-TMTC3 and circ-FAM117B as promising biomarkers for prognosis prediction. What's the presumable function of these two circRNAs in regulating the response to ICB treatment. Is there any evidence that these circRNAs could also bind cancer-related miRNAs?

Response: We thank the reviewer's comment. We predicted circRNAs-miRNA interactions by miranda³². We performed the RT-PCR to examine the top circ-TMTC3 related miRNA, including hsa-miR-34b-5p, hsa-miR-200c-3p, hsa-miR-21-5p, and hsa-miR-142-5p, and circ-FAM117B related miRNA, hsa-miR-142-5p, between the responders and non-responders with ICB treatment (**Fig. A8**). As circ-TMTC3 and circ-FAM117B are highly expressed in non-response group, we noticed significant down-regulation of these miRNAs in non-response group. Furthermore, these miRNAs have been reported to directly target PD-L1³³⁻³⁶, which are associated with immune escape.

Fig. A8. RT-qPCR for relative expression of miRNAs. The relative expression of hsa-miR-34b-5p, hsa-miR-200c-3p, hsa-miR-21-5p, and hsa-miR-142-5p were measured by RT-qPCR in responders (n = 7) and non-responders (n = 7) with ICB-treatment. P value by two-sided Student T-test. *P < 0.05, **P < 0.01, ***P < 0.001.

4. In Fig.3J, Fig.4B, and Fig.4E, the authors used the AUC method to evaluate the performance of the ICBCircSig score. However, the details of calculating the AUC lines are absent in the Methods section. Considering that all ROC curves are smooth curves rather than broken lines, were these results calculated from repeated cross-validation? The authors should provide more details in the Methods section.

Response: The timeROC function in R package timeROC was used to calculate time-dependent receiver-operating characteristic (ROC) curves and area under the ROC curve (AUC) from censored survival data for ICBCirSig score. TimeROC explore a nonparametric estimator for the

time-dependent AUC at each time point, and propose an generalized linear regression for time-dependent AUC regression models³⁷.

For individual ROC curve, we firstly compute the TP and FP value for each time point, which can form ROC curve of broken lines. In order to make the ROC curve smooth, we transformed the TP and FP value to the smooth ROC curves by geom_smooth function of ggplot2 (Fig.3J, Fig.4B, and Fig.4E).

We added a detailed description of AUC in the Method section “*Statistical analysis*” in the revised manuscript (page 18, line 486-488).

Minor comments:

1. “Find_circ” should be written as “find_circ”, which is more commonly used.

Response: We corrected it in updated figures (**Fig. 1a-d, Supplementary Fig. 1c-d**) in the revised manuscript.

2. The confidence interval could be added to the ROC curve if repeated measures are involved.

Response: We performed the time-dependent receiver-operating characteristic (ROC) curves individually and plot each curve together (**Supplementary Fig. 3d-e, Supplementary Fig. 4e-f**). Therefore, there is no confidence interval in the ROC curve.

3. There are many grammar mistakes and misused words. For example, “were download from” (line 315, line326), “ICBcircSig score outperforms and other” (line 229), “a novel biomarkers” (line 300), “the prediction of efficacy” (line 304). The language should be edited by native speakers.

Response: We corrected the grammar mistakes and misused word in the revised manuscript.

4. The authors should provide more technical details on how to identify DE circRNAs and construct the ICBcircSig models, including softwares, parameters, etc. It seems that the authors used a simple statistical model to detect differentially expressed circRNAs, which may generate false positives. A more sophisticated approach CIRIquant should be used or benchmarked here.

Response: Since the length of RNA-seq reads in cohort 2 is 75 bp and the CIRIquant has low detection sensitivity in the sequencing read length less than 100 bp²⁹, we did not use CIRIquant in our original manuscript. We used CIRIquant²⁹ to quantify circRNA in cohort 1. After retained circRNA with at least 2 reads and considered circRNAs identified in more than 20% of the total samples in cohort 1, 4963 circRNAs were identified by CIRIquant, which support the robustness of methods to identify circRNAs ($P < 2.2e-16$ using Fisher’s exact test; **Fig. A9**).

Fig. A9. Overlap of circRNAs identified in this study and CIRIquant

Venn diagram comparison between 5350 circRNAs were retained in cohort 1 based on four circRNA-detection tools and 5900 circRNAs quantified by CIRIquant. P value by Fisher's exact test.

5. Several bioinformatic tools and databases used in this study should be cited. For example, the dataset of PRJEB23709, FastQC, samtools, CIRI2(PMID: 28334140, PMID: 25583365), Kaplan-Meier survival analysis.

Response: We have cited these related datasets and tools in the revised manuscript.

Reviewer #3 (Remarks to the Author): expertise in bioinformatics analysis of transcriptomic data

In this work, authors have identified some key immune-related circRNA in melanoma. Taking advantage of the patients with immune checkpoint blockades treatment, a circRNA-miRNA-mRNA regulatory axes was developed. However, it has not been discussed in depth. Besides, authors have constructed and validated a ICBcircSig signature to distinguish ICB non-benefit patients and benefit patients. There were some issue need to be solved.

Response: We thank the reviewer's positive comments and constructive suggestions. We have experimentally validated the immune-related circRNA in circRNA-miRNA-mRNA regulatory axes (**Please refer to the reviewer #3's comment #3**), we also extensively discussed the clinical application and limitation of ICBcircSig signature in **Discussion** section in the revised manuscript (page 14-15, line 372-384).

Major point:

1. In the method 'Clinical Information and RNA-Seq data of melanoma patients', author gives the clinical information of melanoma patients prior to immunotherapy (PRE) and patients early during treatment. However, patient's response was determined for the PRE. Why the patients without treatment can be classified into responder and non-responder? It's better to explain it more clearly.

Response: We thank the reviewer for this constructive suggestion. The biopsies were obtained prior to immunotherapy (PRE). At the subsequent follow-up, patients were classified into responders and non-responders according to guidelines for The Response Evaluation Criteria in Solid Tumors (RECIST) 1.1 criteria³⁸. We added this in the Method section "**Clinical Information and RNA-Seq data of melanoma patients**" (page 15, line 394-399).

2. How author predict miRNA-circRNA interaction by Miranda? Please give a detailed description, especially for the confidence of predicted targets. Moreover, authors can validate the interactions in transcription level

Response: We predicted circRNAs-miRNA interactions, by miranda³², which can detect potential target sites of miRNAs through complementary sequence information. Briefly, sequence information of each circRNA were obtained by bedtools³⁹, and sequence information of each human miRNA were downloaded from miBase⁴⁰. We then used miranda to predict target sites of miRNAs for each circRNA with following parameters: alignments with energies < -8 and alignments with scores >= 150).

We further performed the RT-PCR to validate the circ-TMTC3 interacted miRNA, including hsa-miR-34b-5p, hsa-miR-200c-3p, hsa-miR-21-5p, and hsa-miR-142-5p, and circ-FAM117B related miRNA hsa-miR-142-5p predicted by Miranda are significantly enriched in responders compared to non-responders. We observed that these miRNAs significantly downregulated in non-responders (**Fig. A10**).

Fig. A10. RT-qPCR for relative expression of miRNAs. The relative expression of hsa-miR-34b-5p, hsa-miR-200c-3p, hsa-miR-21-5p, and hsa-miR-142-5p were measured by RT-qPCR in responders (n = 7) and non-responders (n = 7) with ICB-treatment. P value by two-sided Student T-test. *P < 0.05, **P < 0.01, ***P < 0.001.

3. Functional analysis based on circRNA-miRNA-mRNA regulatory axes needs more explanations for immune correlations. And it is better to prove the relevance of immune regulation through the experiment.

Response: To validate the functional role of circRNA signatures through the circRNA-miRNA-mRNA regulatory axes, we generated the circ-TMTC3 overexpression (OE) cell lines and circ-FAM117B OE cell lines in SK-MEL-28 (**Fig. A11a and d**). According to miranda's prediction³², hsa-miR-142-5p can interact with circ-TMTC3 and circ-FAM117B, we observed hsa-miR-142-5p significantly downregulated in OE cell lines of circ-TMTC3 or circ-FAM117B (**Fig. A11b and e**). Then, we found the expression of PD-L1 is significantly upregulated in both OE cell lines by RT-PCR analysis (**Fig. A11c and f**). These suggest that ICBcircSig signature circ-TMTC3 and circ-FAM117B can regulate the hsa-miR-142-5p/PD-L1 pathway in melanoma cell line.

To further investigate the functional role of ICBcircSig signature circ-TMTC3 and circ-FAM117B in the immunosuppression, we performed an *in vitro* T cell cytotoxicity-mediated tumor killing assay based on SK-MEL-28 melanoma cells overexpressing circ-TMTC3 or circ-FAM117B. The overexpression of circ-TMTC3 or circ-FAM117B significantly reduced the CD8+ T cell cytotoxicity and the ability of these T cells to eliminate tumor cells (**Fig. A11g-i**). Taken together, our results suggest circ-TMTC3 or circ-FAM117B promote the immune escape through hsa-miR-142-5p/PD-L1 pathway. We added these results in the revised manuscript (page 13, line 327-342).

Fig. A11. circ-TMTC3 and circ-FAM117B regulates the miR-142-5p /PD-L1 pathway in the SK-MEL-28 melanoma cell line to enhance immune suppression. (a-c) The expression of circ-TMTC3 (a), miR-142-5p (b), and PD-L1 (c) in SK-MEL-28 melanoma cell line with circ-TMTC3 overexpression (oe) or control group. **(d-f)**The expression of circ-FAM117B (a), miR-142-5p (b), and PD-L1 (c) in SK-MEL-28 melanoma cell line with circ-FAM117B overexpression (oe) or control group. **(g-i)** T cell cytotoxicity-mediated tumor killing assay for SK-MEL-28 melanoma cells with overexpression of circ-TMTC3 or circ-FAM117B. **(g)** SK-MEL-28 melanoma cells with or without overexpression of circ-TMTC3 or circ-FAM117B co-cultured with activated T cells for 48 h were subjected to crystal violet staining. SK-MEL-28-to-T cell ratio, 1:3. **(h-i)** Statistical analysis comparing the ability of T cell killing in (g). Results are mean \pm s.d. Two-side Student T-test was used in **a-f**, one-way ANOVA and Dunnett's multiple comparison test was used in **h-i**. * $p < 0.05$, ** $p < 0.01$ and *** $p < 0.001$.

4. Authors should explain the process of GSEA for accessing the function enrichment in patients with high ICbcircSig score and low ICbcircSig score as detailed as possible.

Response: We thank the reviewer for this comment. We have the detailed description in Method section "**Gene set enrichment analysis**" in the revised manuscript.

5. In this work, author identified the immune-related circRNAs by a series of computational methods. However, the mechanism of these genes mediating immune regulation in ICB treatment has not been discussed. Authors should provide more evidence to support the result, especially experimental verification.

Response: We thank the reviewer for this suggestion. we revealed that the overexpression of ICbcircSig circTMTC3 and circFAM117B could increase PD-L1 expression in cancer via the

miR-142-5p/PD-L1 axis, thus reduce T cell activity and lead to tumor immune escape (**Please refer to the reviewer #3's comment #3**).

Minor point:

1. There is an incorrect uppercase format in the title of method 'Clinical Information and RNA-Seq data of melanoma patients'.

Response: We have corrected it as "Clinical information and RNA-Seq data of melanoma patients".

2. It seems that there is a spelling mistake in line 366 of method 'Identification of circRNA-miRNA-mRNA regulatory axes'.

Response: We have corrected it in the revised manuscript.

3. Please give the detailed parameters during developing the LASSO Cox regression model.

Response: We used *Cv.glmnet* function in R package *glmnet*⁴¹. We first set seed as 123, and deviance to measure loss to use for cross-validation at 5 folds, to develop the LASSO Cox regression model. We then filtered circRNAs with lambda coefficient > 0. We added the description of the detailed parameters for the LASSO Cox regression model in the Method section "***Development of ICBcircSig signature by machine learning***" (page 18, line 470-473).

References

1. Auslander, N. *et al.* Robust prediction of response to immune checkpoint blockade therapy in metastatic melanoma. *Nature Medicine* (2018).
2. Chen, P. L. *et al.* Analysis of immune signatures in longitudinal tumor samples yields insight into biomarkers of response and mechanisms of resistance to immune checkpoint blockade. *Cancer Discov.* (2016) doi:10.1158/2159-8290.CD-15-1545.
3. Hugo, W. *et al.* Genomic and Transcriptomic Features of Response to Anti-PD-1 Therapy in Metastatic Melanoma. *Cell* **165**, 35–44 (2016).
4. Prat, A. *et al.* Immune-related gene expression profiling after PD-1 blockade in non-small cell lung carcinoma, head and neck squamous cell carcinoma, and melanoma. *Cancer Res.* **77**, 3540–3550 (2017).
5. Riaz, N. *et al.* Tumor and Microenvironment Evolution during Immunotherapy with Nivolumab. *Cell* **171**, 934–949 (2017).
6. Weinstein, J. N. *et al.* The cancer genome atlas pan-cancer analysis project. *Nat. Genet.* **45**, 1113–1120 (2013).
7. Westholm, J. O. *et al.* Genome-wide Analysis of Drosophila Circular RNAs Reveals Their Structural and Sequence Properties and Age-Dependent Neural Accumulation. *Cell Rep.* **9**, 1966–80 (2014).
8. Zhang, X. L., Xu, L. L. & Wang, F. Hsa_circ_0020397 regulates colorectal cancer cell viability, apoptosis and invasion by promoting the expression of the miR-138 targets TERT and PD-L1. *Cell Biol. Int.* **41**, 1056–1064 (2017).
9. Huang, Y. *et al.* P53 regulates mesenchymal stem cell-mediated tumor suppression in a tumor microenvironment through immune modulation. *Oncogene* **33**, 3830–3838 (2014).
10. Du, W. W. *et al.* Foxo3 circular RNA retards cell cycle progression via forming ternary complexes with p21 and CDK2. *Nucleic Acids Res.* **44**, 2846–2858 (2016).
11. Manguso, N., Giuliano, A. E. & Tanaka, H. circRNA meets gene amplification. *Non-coding RNA Investig.* **2**, 38 (2018).
12. Babin, L. *et al.* Chromosomal Translocation Formation Is Sufficient to Produce Fusion Circular RNAs Specific to Patient Tumor Cells. *iScience* **5**, 19–29 (2018).
13. Ruan, H. *et al.* Comprehensive characterization of circular RNAs in ~1000 human cancer cell lines. *Genome Med.* **11**, 1–14 (2019).
14. Quan, G. & Li, J. Circular RNAs: Biogenesis, expression and their potential roles in reproduction. *J. Ovarian Res.* **11**, (2018).
15. Xu, Z., Li, P., Fan, L. & Wu, M. The potential role of circRNA in tumor immunity regulation and immunotherapy. *Frontiers in Immunology* vol. 9 (2018).
16. Tumeq, P. C. *et al.* PD-1 blockade induces responses by inhibiting adaptive immune resistance. *Nature* **515**, 568–571 (2014).
17. Nishino, M., Ramaiya, N. H., Hatabu, H. & Hodi, F. S. Monitoring immune-checkpoint blockade: Response evaluation and biomarker development. *Nature Reviews Clinical Oncology* vol. 14 (2017).
18. Huang, A. C. *et al.* T-cell invigoration to tumour burden ratio associated with anti-PD-1 response. *Nature* (2017) doi:10.1038/nature22079.
19. Subbiah, V., Solit, D. B., Chan, T. A. & Kurzrock, R. The FDA approval of pembrolizumab for adult and pediatric patients with tumor mutational burden (TMB) ≥ 10 : a decision centered on empowering patients and their physicians. *Ann. Oncol.* **31**, 1115–1118 (2020).

-
20. Samstein, R. M. *et al.* Tumor mutational load predicts survival after immunotherapy across multiple cancer types. *Nat. Genet.* **51**, 202–206 (2019).
 21. Chung, H. C. *et al.* Pembrolizumab treatment of advanced cervical cancer: Updated results from the phase 2 KEYNOTE-158 study. *J. Clin. Oncol.* **36**, 5522–5522 (2018).
 22. Mehra, R. *et al.* Efficacy and safety of pembrolizumab in recurrent/metastatic head and neck squamous cell carcinoma: Pooled analyses after long-term follow-up in KEYNOTE-012. *Br. J. Cancer* **119**, 153–9 (2018).
 23. Kefford, R. *et al.* Clinical efficacy and correlation with tumor PD-L1 expression in patients (pts) with melanoma (MEL) treated with the anti-PD-1 monoclonal antibody MK-3475. *J. Clin. Oncol.* **15**, 3005–3005 (2014).
 24. Wendel Naumann, R. *et al.* Safety and efficacy of nivolumab monotherapy in recurrent or metastatic cervical, vaginal, or vulvar carcinoma: Results from the phase I/II CheckMate 358 trial. *J. Clin. Oncol.* **37**, 2825–34 (2019).
 25. Chen, D. S. & Mellman, I. Oncology meets immunology: The cancer-immunity cycle. *Immunity* **39**, 1–13 (2013).
 26. Lapuente-Santana, Ó., van Genderen, M., Hilbers, P. A. J., Finotello, F. & Eduati, F. Interpretable systems biomarkers predict response to immune-checkpoint inhibitors. *Patterns* **2**, 100293 (2021).
 27. Jiang, P. *et al.* Signatures of T cell dysfunction and exclusion predict cancer immunotherapy response. *Nature Medicine* (2018). doi:10.1038/s41591-018-0136-1.
 28. Kwon, M. *et al.* Determinants of Response and Intrinsic Resistance to PD-1 Blockade in Microsatellite Instability-High Gastric Cancer. *Cancer Discov.* **02**, candisc.0219.2021 (2021).
 29. Zhang, J., Chen, S., Yang, J. & Zhao, F. Accurate quantification of circular RNAs identifies extensive circular isoform switching events. *Nat. Commun.* **11**, (2020).
 30. Chen, S. *et al.* Widespread and Functional RNA Circularization in Localized Prostate Cancer. *Cell* **176**, 831–43 (2019).
 31. Vo, J. N. *et al.* The Landscape of Circular RNA in Cancer. *Cell* **176**, 869–81 (2019).
 32. John, B. *et al.* Human microRNA targets. *PLoS Biol.* **2**, (2004).
 33. Zhang, J. *et al.* MicroRNA-200c-3p/ZEB2 loop plays a crucial role in the tumor progression of prostate carcinoma. *Ann. Transl. Med.* (2019) doi:10.21037/atm.2019.02.40.
 34. Ankasha, S. J., Shafiee, M. N., Wahab, N. A., Ali, R. A. R. & Mokhtar, N. M. Oncogenic role of mir-200c-3p in high-grade serous ovarian cancer progression via targeting the 3'-untranslated region of dlc1. *Int. J. Environ. Res. Public Health* (2021) doi:10.3390/ijerph18115741.
 35. Dong, L., Chen, F., Fan, Y. & Long, J. MiR-34b-5p inhibits cell proliferation, migration and invasion through targeting ARHGAP1 in breast cancer. *Am. J. Transl. Res.* (2020).
 36. Dong, P. *et al.* Control of PD-L1 expression by miR-140/142/340/383 and oncogenic activation of the OCT4–miR-18a pathway in cervical cancer. *Oncogene* (2018) doi:10.1038/s41388-018-0347-4.
 37. Hung, H. & Chiang, C.-T. Estimation methods for time-dependent AUC models with survival data. *Can. J. Stat.* (2009) doi:10.1002/cjs.10046.
 38. Seymour, L. *et al.* iRECIST: guidelines for response criteria for use in trials testing immunotherapeutics. *Lancet Oncol.* **18**, e143–e152 (2017).
 39. Quinlan, A. R. & Hall, I. M. BEDTools: A flexible suite of utilities for comparing genomic features. *Bioinformatics* (2010) doi:10.1093/bioinformatics/btq033.

-
40. Griffiths-Jones, S., Grocock, R. J., van Dongen, S., Bateman, A. & Enright, A. J. miRBase: microRNA sequences, targets and gene nomenclature. *Nucleic Acids Res.* (2006) doi:10.1093/nar/gkj112.
 41. Friedman, J., Hastie, T. & Tibshirani, R. Regularization paths for generalized linear models via coordinate descent. *J. Stat. Softw.* **33**, 1–22 (2010).

REVIEWER COMMENTS

Reviewer #1 (Remarks to the Author):

The authors have comprehensively addressed my concerns, and the additional data, analyses and discussion have strengthened this manuscript. The manuscript has also much improved in terms of grammar and consistency

Reviewer #2 (Remarks to the Author):

The revised manuscript is much improved, and all my concerns have been properly addressed. I'd like to recommend it to be accepted after the authors make two minor changes.

1. The RNA-seq datasets of the in-house cohort should be submitted to the public database.
2. The two bioinformatic tools or databases (fastqc, circatlas) used in the methods should be cited.

Reviewer #4 (Remarks to the Author):

The authors have improved the description of the statistical and bioinformatic analyses in the revised version and have conducted additional experiments that support circRNA induced regulation through miRNAs.

RESPONSE TO REVIEWERS' COMMENTS:

Reviewer #1

The authors have comprehensively addressed my concerns, and the additional data, analyses and discussion have strengthened this manuscript. The manuscript has also much improved in terms of grammar and consistency.

Response: We thank the reviewer for positive comments on our revised manuscript.

Reviewer #2

The revised manuscript is much improved, and all my concerns have been properly addressed. I'd like to recommend it to be accepted after the authors make two minor changes.

Response: We are very pleased that the reviewer stratified our revised work.

1. The RNA-seq datasets of the in-house cohort should be submitted to the public database.

Response: The raw data of bulk RNA-seq generated in this study were deposited in Genome Sequence Archive (GSA) with accession ID HRA003368 and can be obtained by requesting and following the guidelines for GSA for non-commercial use at <https://ngdc.cncb.ac.cn/gsa-human/request/HRA003368>.

2. The two bioinformatic tools or databases (fastqc, circatlas) used in the methods should be cited.

Response: We have cited these two papers in revised manuscripts.

Reviewer #3

The authors have improved the description of the statistical and bioinformatic analyses in the revised version and have conducted additional experiments that support circRNA induced regulation through miRNAs.

Response: We are very pleased that the reviewer stratified our revised work.

REVIEWERS' COMMENTS

Reviewer #2 (Remarks to the Author):

The authors have addressed all my concerns. I'd like to recommend it to be accepted.